# Brain network dynamics in high-functioning individuals with autism

Takamitsu Watanabe[1] & Geraint Rees[1,2]

Theoretically, autism should be underpinned by aberrant brain dynamics. However, how brain activity changes over time in individuals with autism spectrum disorder (ASD) remains unknown. Here we characterize brain dynamics in autism using an energy-landscape analysis applied to resting-state fMRI data. Whereas neurotypical brain activity frequently transits between two major brain states via an intermediate state, high-functioning adults with ASD show fewer neural transitions due to an unstable intermediate state, and these infrequent transitions predict the severity of autism. Moreover, in contrast to the controls whose IQ is correlated with the neural transition frequency, IQ scores of individuals with ASD are instead predicted by the stability of their brain dynamics. Finally, such brain–behaviour associations are related to functional segregation between brain networks. These findings suggest that atypical functional coordination in the brains of adults with ASD underpins overly stable neural dynamics, which supports both their ASD symptoms and cognitive abilities.

[1] Institute of Cognitive Neuroscience, University College London, 17 Queen Square, London WC1N 3AR, UK. [2] Wellcome Trust Centre for Neuroimaging, University College London, 12 Queen Square, London WC1N 3BG, UK. Correspondence and requests for materials should be addressed to T.W. (email: takamitsu.watanabe@ucl.ac.uk).

Coordinated whole-brain neural dynamics are essential for proper control of functionally different brain systems[1–6], efficient integration of complex and multimodal information[7–9], and smooth adaption to transient daily-life situations[6,10]. Given such integrative roles of macroscopic brain dynamics in our cognitive and neural information processing, it is reasonable to assume that the aberrance of large-scale neural dynamics is a key biological mechanism underlying autism spectrum disorder (ASD)[10,11], which is often explained as impairment of global information processing[12,13].

Despite such neurobiological and aetiological importance, temporal changes in whole-brain neural activity patterns in autism remain understudied[10]. Previous neuroimaging studies examining the brains of individuals with ASD reported aberrant responses in focal brain regions[14–18], atypical functional/anatomical brain network architectures[18–22] and disturbed neural synchronization between specific brain areas[11,23,24]. Although a recent study has found atypical temporal interactions between different brain networks in individuals with autism and associated them with their aberrant behavioural inflexibility[25], how whole-brain neural activity patterns change over time in individuals with ASD is still poorly understood. Thus, the relationships between such brain dynamics and ASD symptoms remain to be identified.

Here we aim to characterize such large-scale brain dynamics in autism and specify the associations between the neural dynamics and autistic behaviours by exploiting energy-landscape analysis[26–28]. We adopt this analysis because it can automatically identify relatively stable and dominant brain activity patterns in high-dimensional neural data without any *a priori* behavioural information, and illustrate brain dynamics as staying in and transitions between such dominant brain states.

This data-driven method reveals that brains of individuals with ASD show fewer neural transitions compared to those of neurotypical controls, and such atypically stable brain dynamics underlie both core symptoms in ASD and general cognitive ability. In addition, we find that the neural dynamics are supported by specific functional coordination between large-scale brain networks.

## Results

**Accuracy of model fitting.** We analysed publicly shared resting-state functional magnetic resonance imaging (fMRI) data[29] from 24 high-functioning adults with ASD and 26 age-/sex-/IQ-matched typically developing (TD) individuals (Table 1). We

used a data set collected at a single site (University of Utah) to avoid adverse effects of multisite recording.

To investigate dynamic coordination between functionally different brain systems, we first prepared a time series of average fMRI signals of seven functional brain networks[30] (Fig. 1a,b). We then binarized the seven network activities, and fitted a pairwize maximum entropy model (MEM)[26,27,31] to them (Supplementary Fig. 1). This model could accurately predict the empirical data in both the TD and ASD groups (accuracy $\geq 96.4\%$, $r_{126} \geq 0.98$, $P < 10^{-5}$ in a test of no correlation; Fig. 1f).

**Dominant brain states.** Next, based on this accurately fitted model, we specified dominant brain states that occurred frequently enough to represent brain activity patterns during rest (Fig. 1c). Technically, we calculated so-called energy values of all the possible brain activity patterns ($2^7$ patterns; Fig. 1g), examined hierarchal relationships between the $2^7$ energy values and systematically searched for dominant brain activity patterns that showed locally minimum energy values and were more likely to be observed than similar activity patterns[26,27] (Fig. 1c). Note that this energy value does not indicate any biological energy; rather, it is a statistical index inversely indicating the appearance probability of each brain activity pattern. For example, activity patterns with lower energy values tend to appear more frequently and should be stable.

We found that the TD and ASD groups had energy landscapes with similar hierarchal structures (Fig. 2a) consisting of the same six locally stable brain activity patterns (local minima A–F; Fig. 2b). In both cohorts, the local minima A and B belonged to the same branch, and the local minima C and D were in another branch. Moreover, compared to these four local minima, energy values of the local minima E and F were relatively high, which suggests that these two local minima were not so dominant and stable as the other four.

On the basis of such hierarchal characteristics and similarity between the two groups, we summarized these six local minima into two major brain states (local minima A and B to the major state #1; local minima C and D to the major state #2) and two minor states (local minima E and F to the minor state #1 and #2, respectively; Fig. 2a).

Such an accurate model fitting and the hierarchal structures of energy landscapes were preserved when we changed the threshold for the binarization of brain activity (Supplementary Fig. 2).

**Table 1 | Demographic data.**

| | TD | ASD | *P* value |
|---|---|---|---|
| Number of participants | 26 | 24 | – |
| Age | 25.3 ± 6.3 (18.2–39.3) years | 25.3 ± 5.5 (18.4–38.9) years | 0.96 |
| Sex | Male | Male | – |
| Handedness | Right | Right | – |
| Full IQ* | 112.6 ± 12.0 (89–131) | 109.9 ± 14.2 (90–134) | 0.47 |
| Verbal IQ | 112.1 ± 12.1 (88–127) | 107.5 ± 14.2 (83–130) | 0.22 |
| Performance IQ | 110.3 ± 10.4 (90–125) | 110.25 ± 16.0 (83–133) | 0.98 |
| Mean head motion | 1.2 ± 0.6 (0.34–2.9) mm | 1.3 ± 0.8 (0.28–2.8) mm | 0.41 |
| ADOS total | 1.2 ± 1.4 (0–4)† | 12.8 ± 3.6 (6–21) | $<10^{-14}$ |
| ADOS social | 0.5 ± 0.7 (0–2)† | 4.8 ± 2.1 (2–13) | 0.001 |
| ADOS communication | 0.7 ± 0.9 (0–3)† | 7.0 ± 2.3 (4–11) | $<10^{-10}$ |
| ADOS RRB | 0† | 1.1 ± 1.1 (0–3) | $<10^{-14}$ |

ADOS, Autism diagnostic observation schedule; ASD, autism spectrum disorder; IQ, intelligence quotient; Max, maximum; Min, minimum; RRB, restricted and repetitive behaviour; TD, typically developing; WASI-III, Wechsler abbreviated scale of intelligence.
Mean ± s.d. (Min–Max).
*IQ was measured based on WAIS-III.
†ADOS scores were given to 16 of the 26 TD. The *P* values for the comparisons of ADOS were based on the 16 TD and 24 ASD individuals.

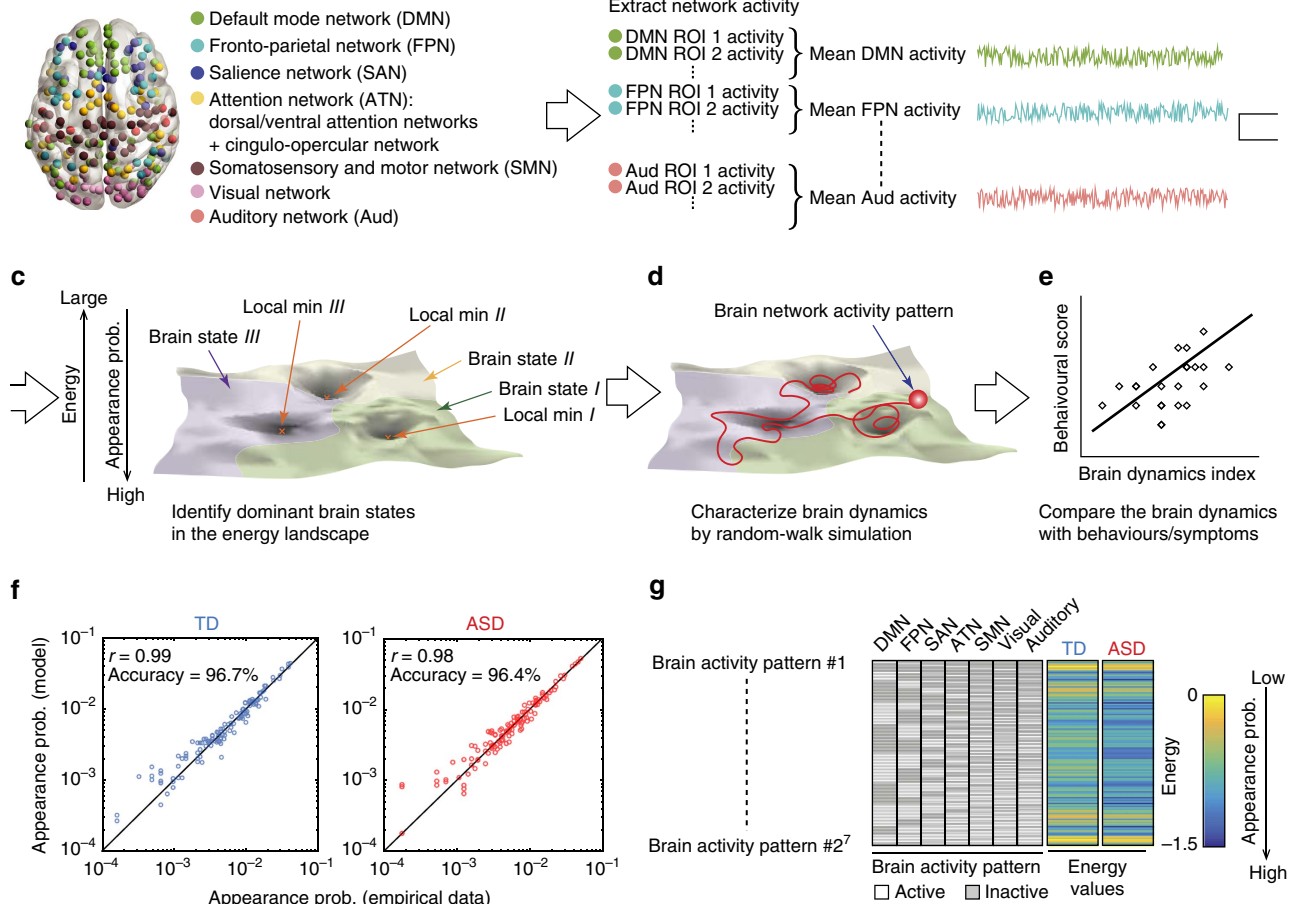

**Figure 1 | Procedures of energy-landscape analysis.** (**a**–**e**) We first extracted resting-state fMRI signals from 214 whole-brain ROIs[30] (**a**), classified the ROIs into seven functionally different brain systems[30] and calculated their average network activity (**b**). By applying a pairwise maximum entropy model to the fMRI data, we built an energy landscape and identified dominant brain states (**c**). After characterizing brain dynamics through random-walk simulation (**d**), we compared several brain dynamic indices with behavioural/symptom scores (**e**). (**f**) In both the TD and ASD groups, the pairwise maximum entropy model showed sufficiently high goodness of fit, and could accurately predict appearance probability (Prob.) of empirical data. (**g**) Accurate model fitting enabled us to accurately infer hypothetical energy values of all the possible $2^7$ brain activity patterns. Note that the energy values do not represent any biological energy, but inversely indicate the appearance probability of the brain activity patterns. That is, a brain activity pattern with a smaller energy value should appear more often. Freq., frequency.

**Sizes of the dominant brain states.** We then quantified the dominance of these major and minor states by calculating how large an area was occupied by each brain state in an energy landscape (that is, the size of 'Brain state' shown in Fig. 1c).

In the TD group, the two minor brain states occupied ~10% of the entire energy landscape (minor state #1: 11.7%, minor state #2: 10.9%), whereas those in the ASD group occupied only ~1.6% (minor state #1: 1.56%, minor state #2: 1.43%). These distributions of the brain state size were significantly different between the two groups ($\chi^3 = 21.9$, $P < 10^{-4}$ in a $\chi^2$-test; Fig. 2c), and the sizes of the minor states were significantly smaller in the ASD group than in the controls ($Z > 3.1$, $P_{uncorrected} < 0.0019$, $P_{Bonferroni} < 0.05$ in *post hoc* residual tests).

Such differences in the brain state size were confirmed by directly counting the appearance frequency of each brain state in the empirical data. The two major brain states appeared more frequently in the ASD group than in the TD group ($t_{48} > 7.8$, $P_{uncorrected} < 10^{-9}$, $P_{Bonferroni} < 0.05$ in two-sample *t*-tests, $P = 0.0001$ in permutation tests, Cohen's $d \geq 2.0$; Fig. 2d), whereas the two minor states showed significantly less appearance

frequency in the ASD group ($t_{48} > 10.6$, $P_{uncorrected} < 10^{-13}$, $P_{Bonferroni} < 0.05$ in two-sample *t*-tests, $P = 0.0001$ in permutation tests, Cohen's $d \geq 2.4$; Fig. 2e).

These results suggest that the minor brain states are atypically unstable and infrequently appeared in individuals with ASD.

**Characterization of brain dynamics.** Next, we performed $10^5$-step random-walk numerical simulation in the energy landscape[26] (Fig. 3a), and characterized brain dynamics as staying in, or transitioning between, these dominant brain states (Fig. 1d).

First, we calculated the transition frequency between the four brain states (Fig. 3b), and found no direct transition between the two minor states for both the TD and ASD groups. Therefore, we categorized neural transitions into either of the following two types of trajectory: a direct transition between the two major brain states or an indirect transition between the two major states via one of the two minor states (Fig. 3c). To simplify descriptions, we hereafter aggregated the two minor brain states into one state called an intermediate state. As with the minor brain states (Fig. 2e), the appearance frequency of this intermediate state was

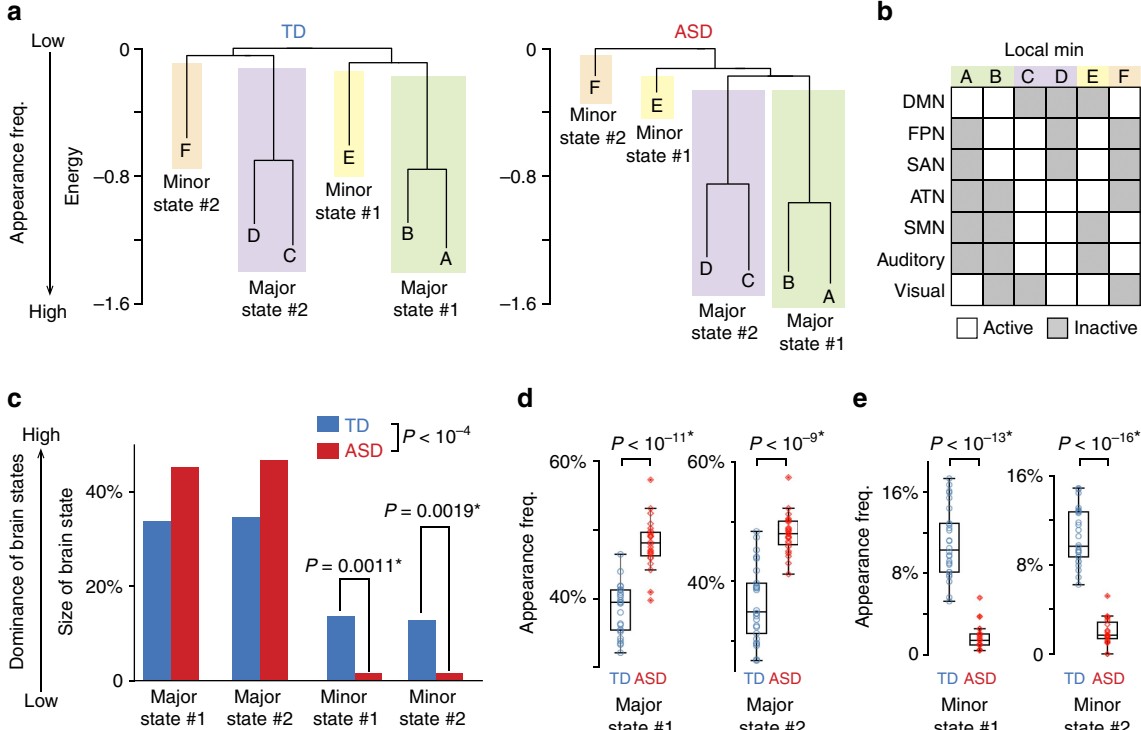

**Figure 2 | Comparison of energy-landscape structures.** (**a,b**) Energy landscapes of the TD and ASD groups showed a similar hierarchal structure (**a**) with the same six local minima (**b**). Given the low energy barriers commonly seen in the two groups, we summarized local minima A and B to the major state #1 and local minima C and D to the major state #2. (**c**) The size of the four dominant states showed significant differences between the TD and ASD groups. In particular, the two minor brain states were significantly smaller in the ASD individuals than in the controls. *$P_{Bonferroni} < 10^{-5}$. (**d,e**) Such differences in brain state sizes were confirmed in the empirical data. In the ASD group, the appearance frequency of the two major states was larger (**d**) and that of the two minor states was smaller (**e**) than in the TD group. *$P_{Bonferroni} < 10^{-5}$.

significantly smaller in the ASD group than in the controls ($t_{48} = 20.3$, $P_{uncorrected} < 10^{-5}$ in a two-sample $t$-test, $P = 0.0002$ in a permutation test, Cohen's $d = 3.5$; Fig. 3d).

We then compared the frequency of these direct and indirect transitions between the TD and ASD groups. The direct transition frequency was not different between the two groups, whereas the indirect transition frequency was significantly reduced in the ASD individuals compared to the controls ($\chi^2 = 283.6$, $P < 10^{-5}$ in a $\chi^2$-test; $Z = 16.8$, $P_{uncorrected} < 10^{-5}$, $P_{Bonferroni} < 0.05$ in a *post hoc* residual test; Fig. 3e). This contrast was reproduced even when we counted each transition frequency in the empirical fMRI data (Fig. 3f): no significant difference was seen in the direct transition frequency ($t_{48} = 1.5$, $P = 0.13$ in a two-sample $t$-test, $P = 0.14$ in a permutation test, Cohen's $d = 0.17$), whereas the indirect transition frequency was significantly lower in the ASD group ($t_{48} = 14.0$, $P_{uncorrected} < 10^{-5}$, $P_{Bonferroni} < 0.05$ in a two-sample $t$-test, $P = 0.0001$ in a permutation test, Cohen's $d = 5.0$).

Such atypically infrequent neural transitions in ASD individuals imply that their brain dynamics are more stable than those of the controls, and thus their brain activity tends to stay in the major states longer. This implication was confirmed by calculating how long a brain activity pattern stayed in either of the two major states. In this random-walk simulation, the ASD brains showed significantly longer duration of the major states than TD brains ($t_{8735} = 3.9$, $P < 10^{-4}$ in a two-sample $t$-test, $P = 0.0001$ in a permutation test, Cohen's $d = 3.1$; Fig. 3g). This difference was reproduced in direct counting of the repetition length of the major states in the empirical data ($t_{48} = 3.6$, $P = 0.0008$ in a two-sample $t$-test, $P = 0.0013$ in a permutation test, Cohen's $d = 1.0$; Fig. 3h).

Finally, we examined hierarchical relationships between the three brain dynamics indices that were sensitive to autism (that is, the indirect transition frequency, intermediate-state frequency and duration of the major states). Partial correlation analyses showed that in both the TD (Fig. 3i) and ASD groups (Fig. 3j) the intermediate-state frequency was positively associated with the indirect transitions (rho > 0.57, $P < 0.0044$, df ≥ 22 in tests of no correlation), and the indirect transition frequency was inversely correlated with the duration of the major states (rho < −0.38, $P < 0.042$, df ≥ 22 in tests of no correlation).

Taken together, these analyses suggest that in the brains of individuals with ASD, atypically unstable intermediate states were related to a reduction in the indirect transition frequency, which was associated with aberrantly long durations of the major brain states (Fig. 3j). In both the TD and ASD groups, these three brain dynamic indices were not significantly correlated with the ages of the individuals (|r| ≤ 0.18, $P ≥ 0.37$, df ≥ 22 in tests of no correlation; Supplementary Table 1).

**Brain dynamics and symptoms of autism.** We then examined whether these indices of brain dynamics were related to symptoms of autism, as measured by the Autism Diagnostic Observation Schedule (ADOS)[32].

In the ASD group, the indirect transition frequency was negatively correlated with ADOS total scores ($r_{22} = −0.47$, $P_{uncorrected} = 0.01$, $P_{Bonferroni} < 0.05$ in a test of no correlation; Fig. 4a), whereas the duration of the major states did not show a significant correlation ($r = −0.09$). Even in the TD group, the indirect transition frequency was significantly smaller in the individuals with higher ADOS scores (ADOS total = 2–4) than in

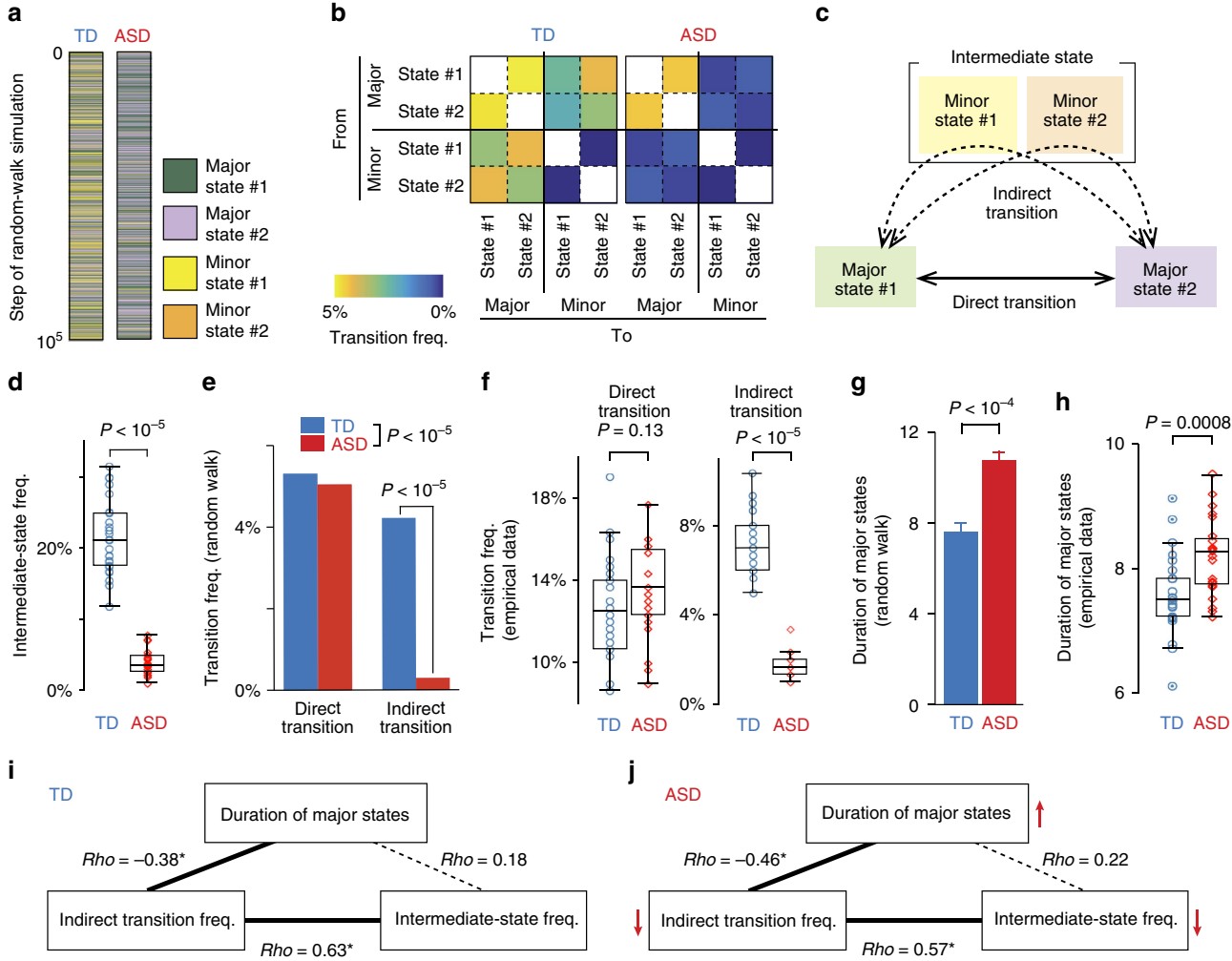

**Figure 3 | Brain dynamics.** (**a–d**) To characterize brain dynamics in the energy landscape, we performed $10^5$-step random-walk simulation (**a**). On the basis of this simulation, we first calculated the transition frequency between the four dominant brain states (**b**), and found no transition between minor states #1 and #2. Therefore, we could classify all transitions into direct transitions between the two major states or indirect transitions via one of the minor states (**c**). On the basis of such classification, we summarized the two minor states into the intermediate state. In the empirical data, the intermediate state was seen significantly less frequently in the ASD group compared to the controls (**d**). (**e,f**) In the simulation, the direct transition frequency was not significantly different between the TD and ASD groups, but the indirect transition frequency was significantly smaller in the ASD individuals (**e**). This contrast was reproduced even by direct counting of frequency of each type of transition in the empirical data (**f**). $*P_{\text{Bonferroni-corrected}} < 0.05$. (**g,h**) The duration of staying in the major states was significantly longer in the ASD group (**g**), which was confirmed in direct counting of the empirical data (**h**). Error bar, s.d. (**i,j**). In both the TD and ASD groups, we found a negative partial correlation between the duration of the major states and the indirect transition frequency, a positive partial correlation between indirect transition frequency and the intermediate-state frequency. We could not detect a significant partial correlation between duration of the major states and the intermediate-state frequency.

those with lower ADOS scores (ADOS total = 0–1; $t_{16} = 2.6$, $P = 0.019$ in a two-sample $t$-test; Supplementary Fig. 3), while the duration of the major states was not significantly different between the TD individuals with higher and lower ADOS scores ($P > 0.56$). This brain–symptom association was not specific to either of the social or non-social core symptoms of autism (Supplementary Fig. 4).

In addition, a partial correlation analysis revealed a hierarchical relationship between ADOS scores, the indirect transition frequency and the appearance frequency of the two local minima in the intermediate state (Fig. 4b).

These observations indicate that the atypically unstable intermediate state in the brains of individuals with ASD is related to the reduction in the indirect transitions, and such aberrant decreases in brain dynamics flexibility are associated with the severity of ASD symptoms.

**Brain dynamics and general cognitive ability.** Second, we examined associations between brain dynamics and general cognitive ability that was measured as full intelligence quotient (FIQ) scores[33–35].

In the TD group, FIQ scores were specifically correlated with the indirect transition frequency ($r_{24} = 0.46$, $P_{\text{uncorrected}} = 0.014$, $P_{\text{Bonferroni}} < 0.05$ in a test of no correlation; Fig. 4c). In addition, a partial correlation analysis identified a hierarchal relationship between FIQ, the indirect transition frequency and the appearance frequency of the two local minima constituting the intermediate state (Fig. 4d). These results suggest that in TD individuals the stability of the intermediate state is linked with an increase in the indirect transitions, which in turn is associated with their general cognitive skills (Fig. 5a).

In contrast, such an association between the indirect transition frequency and FIQ was not found in the ASD group ($r_{22} = -0.25$,

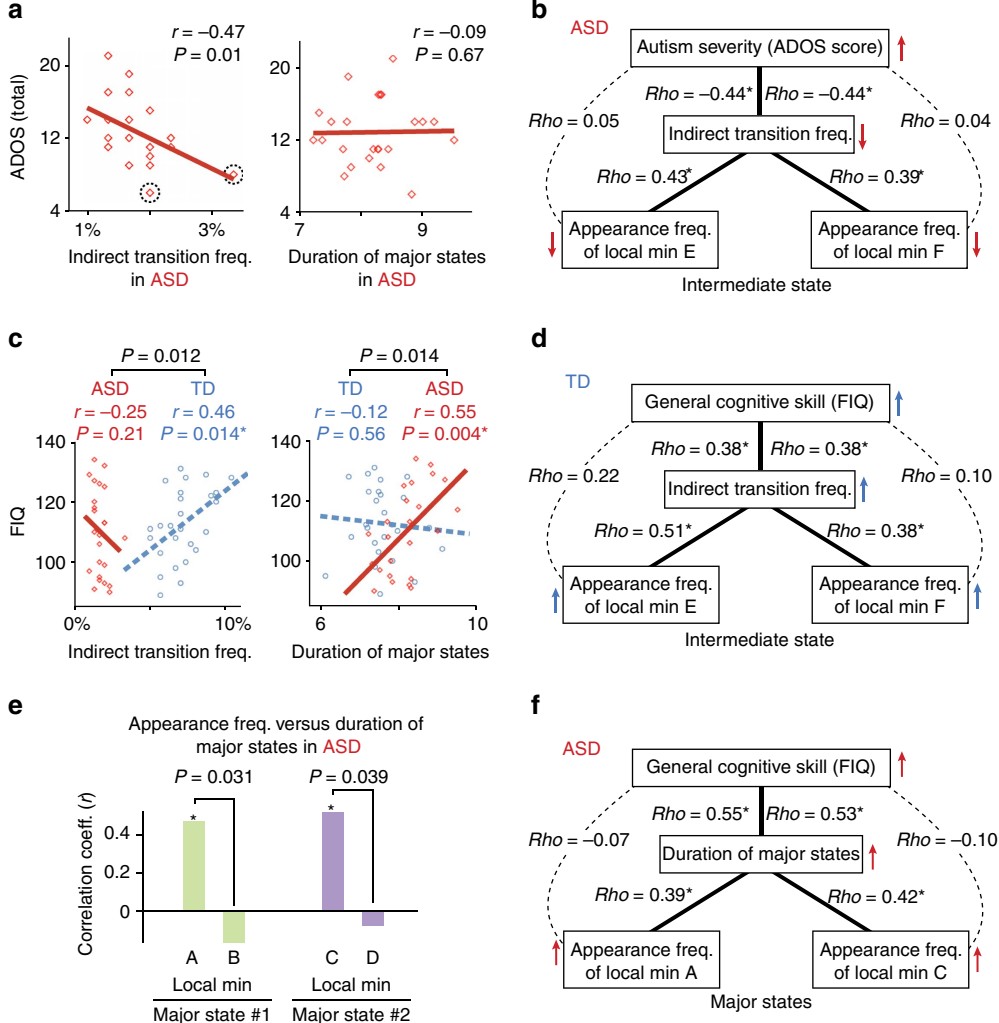

**Figure 4 | Associations between brain dynamics and behaviour.** (**a,b**) In the ASD group, only the indirect transition frequency showed a significant correlation with severity of ASD symptoms ($P_{Bonferroni-corrected} < 0.05$). (**a**) This correlation was preserved even after two outliers (squares circled by dashed lines) were removed ($r = -0.46$). In addition, by partial correlation analysis, we found hierarchal structures between the ADOS total scores, the indirect transition frequency and the appearance frequency of the two local minima constituting the intermediate state (**b**), which suggests that the atypically unstable intermediate state is related to the infrequent indirect transition, and could be associated with the ASD symptoms. *$P < 0.05$. (**c–f**) In the TD group, only the indirect transition frequency was correlated with their general cognitive ability (FIQ; $P_{Bonferroni-corrected} < 0.05$, **c**). Partial correlation analysis showed a hierarchy between FIQ, the indirect transition frequency and the appearance frequency of the local minima in the intermediate state (**d**), which suggests that the stability of the intermediate state is related to increases in the indirect transition and enhancement of their cognitive skills. In contrast, FIQ of the individuals with ASD was correlated with duration of the major states (**c**). In addition, the duration of the major state was associated with the appearance frequency of two of the four local minima in the major states (**e**). These findings and results of partial correlation analysis indicate that the overly stable major states are associated with atypically long duration of the major states in the brains of individuals with ASD, which supports their cognitive ability (**f**). *$P < 0.05$.

$P = 0.21$ in a test of no correlation; left panel of Fig. 4c). Instead, their FIQ scores were correlated with the duration of the major states ($r_{22} = 0.55$, $P_{uncorrected} = 0.004$, $P_{Bonferroni} < 0.05$ in a test of no correlation; right panel of Fig. 4c), which was mainly determined by the appearance frequency of two of the four local minima constituting the major states (local minimum A, $r_{22} = 0.45$, $P_{uncorrected} = 0.024$; local minimum C, $r_{22} = 0.49$, $P_{uncorrected} = 0.012$; both, $P_{Bonferroni} < 0.05$ within each major state in tests of no correlation; Fig. 4e).

Taken together with the results of the partial correlation analysis (Fig. 4f), these observations indicate that the general cognitive ability of ASD individuals is associated with the stability of their brain dynamics, which in turn are supported by the atypically stable major brain states represented by the two specific local minima (Fig. 5b).

**Brian network coordination related to symptoms of autism.** We then examined across-network functional coordination underlying these associations between the stability/flexibility of brain dynamics and ADOS/IQ scores.

First, we investigated across-network functional connectivity (FC) underpinning the atypical reduction of the indirect transition frequency (Fig. 6).

As shown in Fig. 4b, the larger frequency of the indirect transition was associated with the higher appearance probability of the two local minima constituting the intermediate state. On the other hand, these local minima show complementary brain activity patterns, and are based on anticorrelated activation of two network modules (that is, Default-mode/Sensory-motor/Auditory (DMN/SMN/Auditory) module and Fronto-parietal/Salience/Attention/Visual (FPN/SAN/ATN/Visual) module; Fig. 6a). Given these

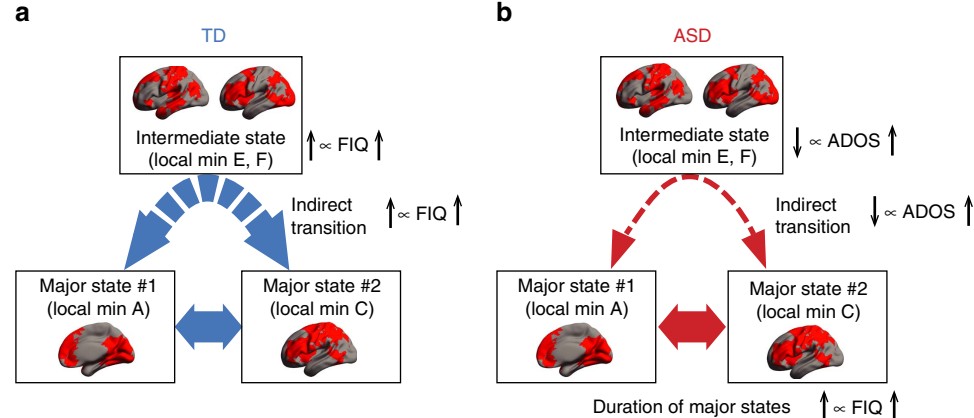

**Figure 5 | Brain dynamics and behaviour.** In the TD individuals (**a**), the flexibility of their brain dynamics was correlated with their cognitive ability. In the ASD individuals (**b**), their atypical stability of brain dynamics was associated with both their ASD symptoms and cognitive skills. The brain maps represent binary brain activity patterns of the dominant brain states. Red areas were active regions (+1), whereas the other areas were inactive regions (–1).

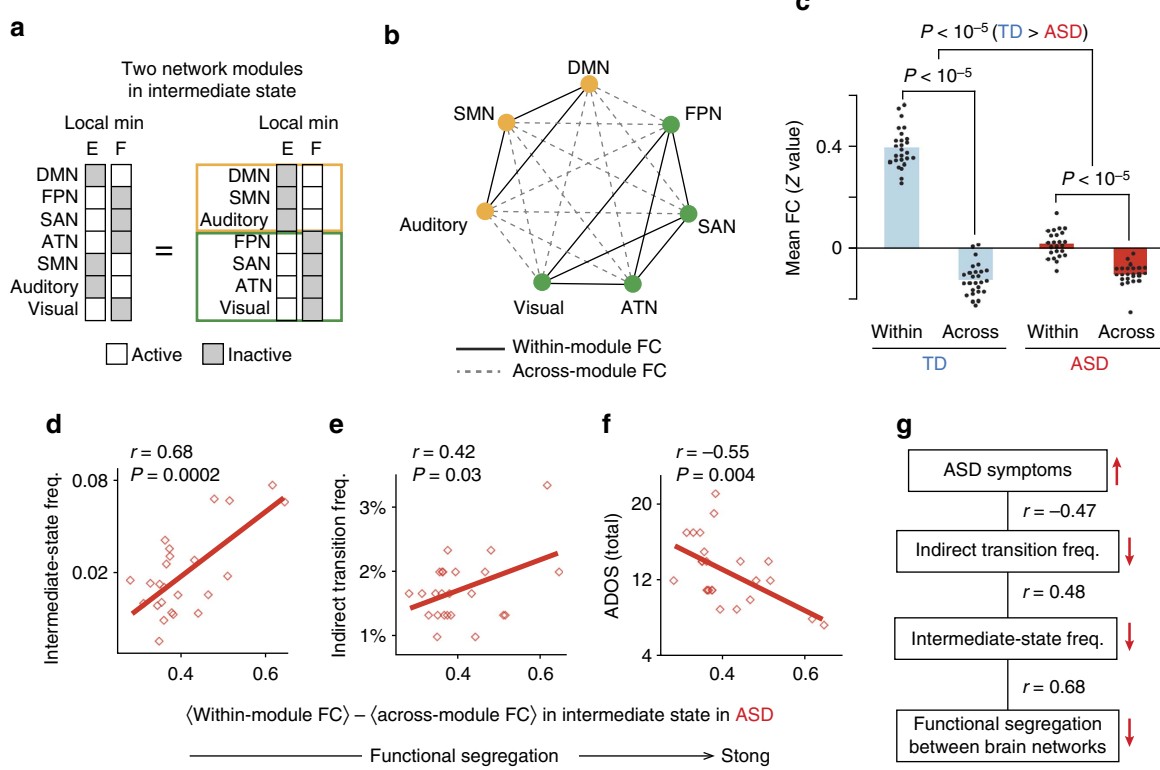

**Figure 6 | Across-network functional segregation during intermediate state in autism.** (**a–c**) The intermediate state was based on two modules that were complementary and anticorrelated with each other (DMN/SMN/Auditory module and FPN/SAN/ATN/Visual module; **a**). Thus, it is reasonable to assume that strong functional segregation between these modules should stabilize the intermediate state. To examine this hypothesis, we first quantified the functional segregation strength by comparing within-module FC with across-module FC (**b**), and found the functional segregation was significantly stronger in the TD group (**c**). (**d–g**) In the ASD group, the functional segregation strength was positively correlated with the intermediate-state frequency (**d**) and indirect transition frequency (**e**), and showed a negative association with ADOS total scores (**f**). Atypically weak across-network functional segregation may destabilize the intermediate state and reduce the indirect transition frequency, which might result in ASD symptoms (**g**).

properties, we assumed that in ASD individuals atypically weak segregation between the two network modules should be related to the reduced appearance frequency of the intermediate state and the decrease in the indirect transition frequency, which is consequently associated with the deterioration of the ASD symptoms.

We tested this hypothesis by comparing within-module FCs to across-module FCs (Fig. 6b) because, theoretically, the two network modules should become more functionally segregated as the gap between within-module and across-module FCs expands[8,36].

In both the TD and ASD groups, the within-module FCs were significantly larger than the across-module FCs ($F_{1,96} = 690.9$, $P < 10^{-5}$ as a main effect of FC types in a two-way factorial analysis of variance (ANOVA) with an (FC type: within/across-module) × (Group: TD/ASD) structure; Fig. 6c).

However, the gap between the within- and across-module FCs was significantly smaller in the ASD group compared to the controls ($F_{1,96} = 272.2$, $P < 10^{-5}$ as an interaction in the two-way factorial ANOVA; $t_{48} = 13.1$, $P < 10^{-5}$ in a *post hoc* two-sample

*t*-test, $P = 0.0001$ in a *post hoc* permutation test, Cohen's $d = 3.0$; Fig. 6c), which suggests that the brains of ASD individuals have weaker functional segregation between the two brain modules.

This atypically weaker functional segregation was correlated with the intermediate-state frequency ($r_{22} = 0.68$, $P = 0.0002$ in a test of no correlation; Fig. 6d) and the indirect transition frequency ($r_{22} = 0.42$, $P = 0.03$ in a test of no correlation; Fig. 6e). Moreover, this functional segregation strength was inversely associated with ADOS total scores ($r_{22} = -0.55$, $P = 0.004$ in a test of no correlation; Fig. 6f). Such a significant association between ADOS scores and the functional segregation strength was observed even in the TD data ($t_{16} = 2.5$, $P = 0.025$ in a two-sample *t*-test, $P = 0.0001$ in a *post hoc* permutation test, Cohen's $d = 1.4$; Supplementary Fig. 5a).

Considering these results with our other observations (Figs 4b and 5b), the current findings indicate that aberrantly weak functional segregation between specific brain modules is related to the unstable intermediate state and fewer indirect transitions in individuals with ASD, which stabilizes their brain dynamics and in turn underlies the symptoms (Fig. 6g).

**Brian network coordination related to cognitive ability**. Next, we examined across-network functional coordination that could be related to the atypically long duration of the major states in the individuals with ASD and underlie their general cognitive ability (Fig. 7).

On the basis of the observation that the overly stable brain dynamics in individuals with ASD were associated with the aberrantly large appearance frequency of two complementary local minima (Figs 4e and 7a), we assumed that these stable dynamics were related to functional segregations between two different network modules (DMN/Visual module and FPN/SAN/ATN/SMN/Auditory module; Fig. 7b).

Although significant functional segregation was seen in both the TD and ASD groups ($F_{1,96} = 72.3$, $P < 10^{-5}$ as a main effect of FC types in a two-way factorial ANOVA; Fig. 7c), its strength was significantly larger in the ASD individuals ($F_{1,96} = 5.6$, $P = 0.020$ as an interaction in a two-way factorial ANOVA; $t_{48} = 2.3$, $P = 0.02$ in a *post hoc* two-sample *t*-test, $P = 0.03$ in a *post hoc* permutation test, Cohen's $d = 0.63$; Fig. 7c). Moreover, this atypically strong functional segregation was positively correlated with the major state frequency ($r_{22} = 0.61$, $P = 0.001$ in a test of no correlation; Fig. 7d), the duration of the major states ($r_{22} = 0.42$, $P = 0.03$ in a test of no correlation; Fig. 7e) and FIQ ($r_{22} = 0.51$, $P = 0.009$ in a test of no correlation; Fig. 7f) in autism. In contrast, the correlation between FIQ and the functional segregation during the major states was not positive but significantly negative in the TD data ($r_{24} = -0.44$, $P = 0.023$; Supplementary Fig. 5b), suggesting that high-functioning individuals with ASD have different cognitive styles compared to TD individuals[37–39].

These results indicate that atypically strong functional segregation in the brains of individuals with ASD is associated with the aberrantly stable major brain states and atypically long duration of staying in the states, which is consequently correlated with their general cognitive skills (Fig. 7g).

According to the same logic, we confirmed that in the TD group strong functional segregation between DMN/SMN/Auditory module and FPN/SAN/ATN/Visual module (Figs 6c and 8a) was associated with the large appearance frequency of the intermediate state and the indirect transition, which in turn was related to their high IQ scores (Fig. 8b,c).

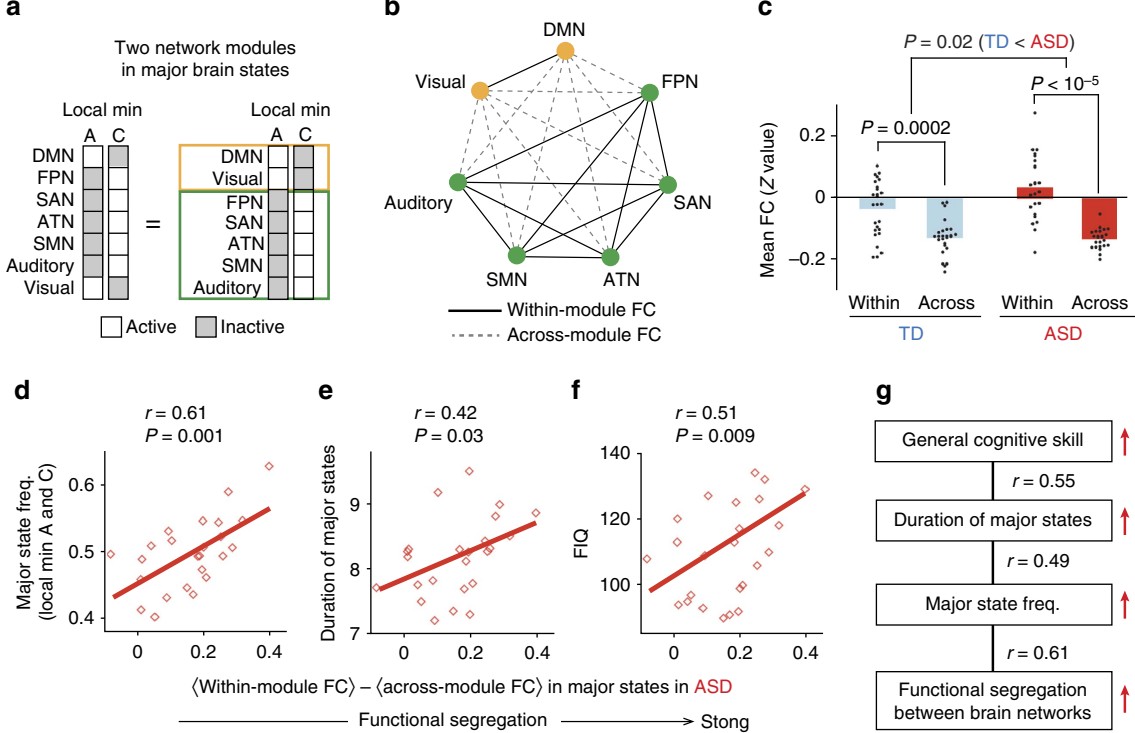

**Figure 7 | Across-network functional segregation during major brain states in autism.** (**a–c**) We assumed that the stability of the major states should be underpinned by functional segregation between a DMN/Visual module and an FPN/SAN/ATN/SMN/Auditory module (**a,b**). The across-network functional segregation was significantly stronger in the ASD group than in the controls (**c**). (**d–g**) In the ASD group, the atypically strong functional segregation was correlated with the major state frequency (**d**), the duration of the major states (**e**) and the FIQ scores (**f**). These results imply that aberrantly strong functional segregation in autism overly may stabilize brain dynamics, which would be related to their cognitive skills (**g**).

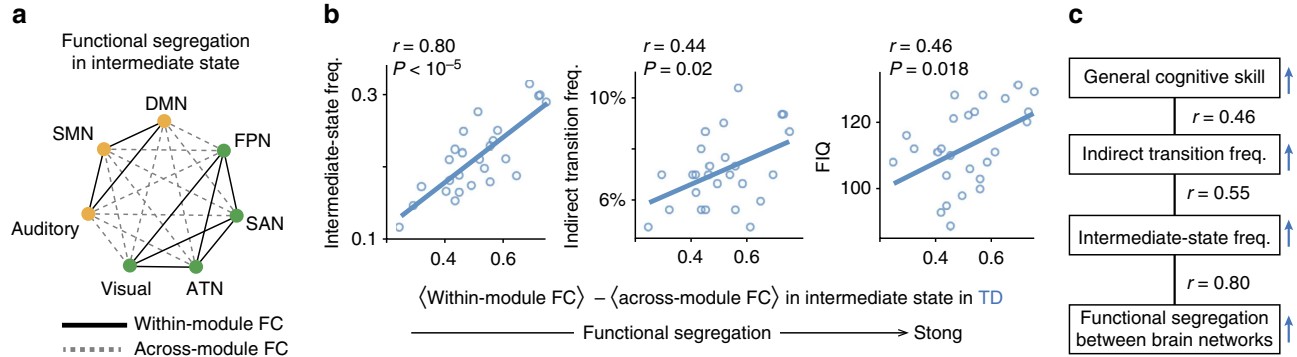

**Figure 8 | Associations between across-network functional coordination and brain dynamics in TD.** We confirmed that such across-network functional segregation underlay the brain dynamics of neurotypical individuals. Here we calculated the strength of functional segregation related to the intermediate state (**a**), and showed significant correlations between the functional segregation strength and (i) the intermediate-state frequency, (ii) the indirect transition frequency and (iii) FIQ scores (**b**). These results suggest that in TD individuals functional segregation between the specific two network modules may stabilize the intermediate state and ensure the flexibility of brain dynamics, which would be associated with general cognitive ability (**c**).

**Reproducibility tests**. We confirmed the robustness of the findings using two independent data sets collected in Indiana University (Supplementary Table 2) and ETH Zürich (Supplementary Table 3). In both data sets, a pairwise MEM was accurately fitted (Supplementary Figs 6a and 8a), and energy-landscape analyses yielded qualitatively the same hierarchal structures of the energy landscapes, consisting of the same six brain states (Supplementary Figs 6b and 8b). In addition, significant differences in the major/intermediate-state frequency between TD and ASD groups were also reproduced (Supplementary Figs 6c and 8c). Moreover, we could identify the atypically lower indirect transition frequency and aberrantly longer duration of the major states in the ASD groups (Supplementary Figs 6d–g and 8d–g). Finally, we confirmed that the correlations between brain dynamics and behavioural indices were reproduced (Supplementary Figs 6h–j and 8h–j), and the associations between these brain dynamics and across-network functional coordination were also replicated (Supplementary Figs 7 and 9).

We also tested whether the current observations were robust across different definitions of brain networks. To this end, we repeated the energy-landscape analyses after the brain was parcellated in two different manners[40,41].

In one of the brain parcellation methods, the DMN was divided into two subnetworks according to a previous study[40] (see Methods for details). Although the accuracy of the model fitting was slightly reduced (82.1% for TD and 80.7% for ASD), we observed qualitatively the same energy landscapes, brain dynamics and brain–behaviour associations as seen in the original analyses (Supplementary Fig. 10).

In the other brain parcellation method, the cortical area was divided into a different set of seven brain systems based on another previous study[41] (Supplementary Fig. 11a). Even with this brain parcellation scheme, we observed qualitatively the energy-landscape structures (Supplementary Fig. 11b), and found that the neural dynamics of individuals with ASD were more stable than those of the control (Supplementary Fig. 11c–g). This neural stability showed positive correlations with the severity of tASD symptoms and general cognitive ability (Supplementary Fig. 11h–j). These brain–behaviour associations were attributable to the strength of functional segregation between the seven brain systems (Supplementary Fig. 11k–n).

**Prediction of diagnosis of autism**. We then examined whether such differences in brain dynamics can predict the diagnosis of autism (Fig. 9 and Supplementary Fig. 12). We first defined a threshold for the diagnosis using the original data set (University

of Utah data), and then evaluated the performance of this diagnostic approach by applying it to the independent data sets (Indiana University and ETH Zürich data). For the test data, the energy-landscape analysis was performed not at a group but at an individual level, and, therefore, the results were not exactly the same as those obtained in the above reproducibility test.

The intermediate-state frequency could predict the ASD diagnosis with relatively high accuracy (sensitivity = 84%, specificity = 85%; Fig. 9a). In contrast, the indirect transition frequency did not realize such accurate prediction (sensitivity = 68%, specificity = 74%; Fig. 9b). We could improve the diagnosis accuracy by combining the two indices in a multivariate pattern analysis method (sensitivity = 89%, specificity = 93%; Fig. 9c,d). This classification accuracy is comparable to or higher than the diagnostic accuracy reported by previous resting-state fMRI studies (for example, sensitivity = 67–83%, specificity = 75–100%)[42–44].

## Discussion

Using energy-landscape analysis, we elucidated atypical resting-state brain dynamics underlying the symptoms and general cognitive ability of high-functioning ASD individuals. Brain dynamics seen in the TD and ASD groups commonly consisted of direct transitions between the two major states and indirect transitions via the intermediate state. However, the indirect transition in the ASD group was aberrantly infrequent because of their unstable intermediate state, and their brain activity was likely to stay in the major states for atypically long durations. Such aberrant reduction in the indirect transitions was related to the severity of autism, whereas the long duration of the major states was correlated with IQ scores in the ASD group. Moreover, the overly stable brain dynamics of the individuals with ASD were linked to aberrant coordination between functionally different brain systems. These findings indicate that, in high-functioning ASD adults, the atypical balance of large-scale brain systems is associated with aberrantly stable brain dynamics, which underlies both their ASD symptoms and general cognitive ability.

This study provides empirical evidence for the concept that autism can be characterized by atypical large-scale brain dynamics[11]. Although previous human neuroimaging studies reported the disturbance of neural synchrony in individuals with ASD[24,45,46] and identified a variety of structural/functional whole-brain architectures specific to ASD[18–20,47], most of them did not directly investigate brain dynamics. Exceptionally, a recent fMRI study using Granger causality analysis investigated patterns of temporal interactions between different brain regions

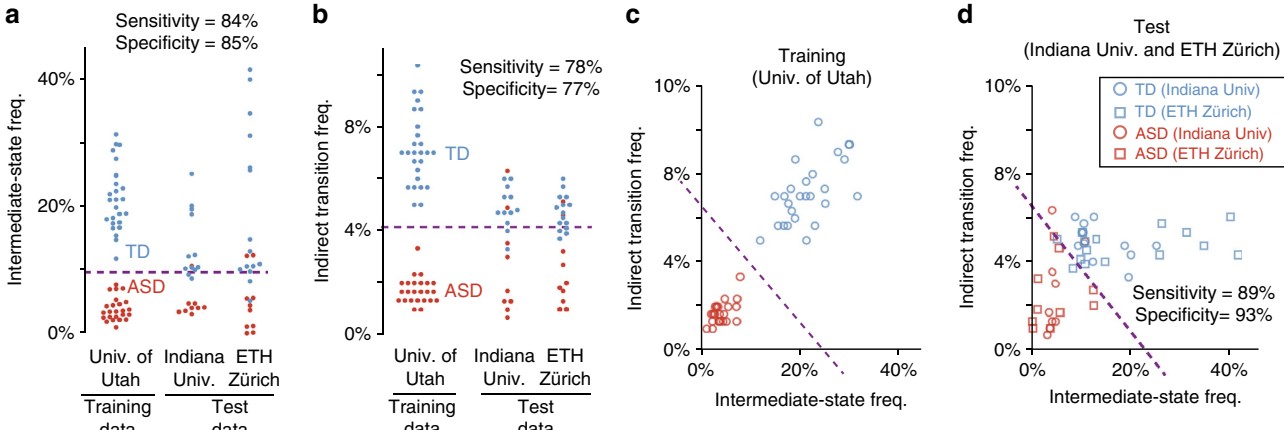

**Figure 9 | Diagnosis prediction.** The thresholds for the diagnostic prediction were defined in the training data, and subsequently evaluated in the independent test data (Supplementary Fig. 10). The two indices of brain dynamics for the test data were calculated for each participant. Therefore, the accuracy of model fit in the energy-landscape analysis (*ca.* 80%) was not so high as in the group-level analysis (*ca.* 95%). (**a**,**b**) The cutoff values for the univariate diagnosis predictions were 7.7% for the intermediate-state frequency (**a**) and 4.15% for the indirect transition frequency (**b**). (**c**,**d**). The cutoff line for the bivariate prediction (purple line; **c**) was defined by training a linear support vector machine with the University of Utah data set. This line could classify individuals in the independent data sets with a relatively high accuracy (**d**). Univ., University.

and reported atypically stable temporal changes in FC[25]. However, even this study did not examine how neural states stay in and transit between different activity patterns over time. In contrast, the current study has characterized such temporal changes in whole-brain activity patterns, and directly demonstrated the associations between atypical brain dynamics seen in ASD individuals and their symptoms.

Such a critical link between symptoms and brain dynamics is not limited to autism, but has been reported in recent human fMRI studies on schizophrenia[48,49]. For example, patients with schizophrenia and healthy controls showed similar static brain states, but exhibited significant differences in the dwell time in specific brain states and transition frequencies between such brain states[48]. Given such prior observations, the current study can be seen as additional empirical support that highlights the importance of investigating brain dynamics in biological understanding of various developmental and psychiatric disorders[11,50].

The atypical across-network functional coordination seen in the ASD group is consistent with previous observations of atypical across-region FCs in autism[51–53]. A previous resting-state fMRI study reported atypical reduction in FC between the amygdala, which is often included in SAN, and secondary visual area[51]. If this observation implies a weak FC between SAN and the visual network, it matches the current findings about weak segregation during the intermediate state and strong segregation during the major states in autism. According to the same logic, the current results are consistent with another resting-state fMRI study[52] reporting an atypical decrease in the FC between a temporal region (auditory network) and a medial prefrontal area (DMN) in autism. In addition, our results are consistent with a task-based fMRI study that found weak functional coupling between a visual area and a region in FPN in high-functioning ASD adults[53].

The current study has also identified brain dynamics associated with the general cognitive ability in neurotypical adults. The general cognitive ability in the TD participants was positively associated with the flexibility of the brain dynamics, and such flexible brain dynamics were underpinned by the increased functional segregation during the intermediate state and the decreased functional segregation during the major states. This functional coordination during the major and intermediate states

may enable the control of diverse cognitive functions in an integrative manner. Theoretically, smooth integration of functionally different brain systems should be vital for binding diverse perceptual information and achieving better cognitive performance in a changing environment[4,6,36,54]. Empirically, several neuroimaging studies have suggested that such an integration process is achieved by frequent transitions between different brain states[8,55]. Considering the current results with these theoretical and empirical observations, we can speculate that the functional coordination seen in the TD group may contribute to integrative information processing by facilitating transitions between different brain activity states.

In contrast, the general cognitive ability in autism was associated with the stability of the brain dynamics, not with the flexibility, which could fit the unique cognitive style of high-functioning ASD individuals[37–39]. Behaviourally, high-functioning individuals with ASD are likely to show above-average performance when tasks they are engaged in require detail-focused rather than global information processing[12,56]. This behavioural tendency matches the overly stable brain dynamics observed in this study, if, as suggested in a previous study[8], the stability of brain dynamics is related to one's capability of repeating the same cognitive process. The current findings may become a new foundation for a biological understanding of autism-specific cognitive styles.

One of the limitations of this study is that we did not examine associations between brain dynamics and subcategories of the general cognitive ability. This was because the data set did not have other cognitive performance scores other than IQ. Although IQ has been used as a valid index of human general cognitive ability[34–36], it would be necessary to examine brain dynamics underlying more specific cognitive skills by employing appropriate psychological tasks.

We also need to be careful not to conclude that the aberrantly stable brain dynamics of autism are related to every aspect of the disorder. Both the social and non-social core symptoms of ASD showed similar effect sizes for brain–symptom associations in this study, but non-social symptoms could be more relevant to such neurophysiological inflexibility[10,25]. To clarify this issue, it would be necessary to apply the current energy-landscape analysis to task-based neuroimaging data of individuals with ASD.

Another limitation of our work concerns potential heterogeneity in the ASD group[57]. We focused on high-functioning, right-handed male adults without any psychiatric comorbidity including attention deficit hyperactivity disorder (ADHD), but this approach could not control all between-participant differences. In fact, some recent studies have reported that neural responses in autism could be affected by genetic patterns[58,59], and other studies have shown significant diversity in executive function within ASD individuals[60]. Moreover, such heterogeneity could be larger in the ASD group than in the controls[57]. Consistent with this, we observed more outliers in our ASD data than the control data: the ASD data had seven outliers ( > or < mean ± 2 s.d.) in the appearance frequencies of the major/minor brain states, whereas the TD data had no outliers. Therefore, the current observations will need to be further examined in more genetically and behaviourally homogeneous subgroups of ASD.

Our analytic approach also had some methodological limitations. We classified cortical regions into the seven systems, and examined system-level brain dynamics. Although similar approximations have yielded biologically meaningful observations in other neuroimaging studies[2,3,5,7], such system-level summarization of brain signals may lose detailed and nuanced information that should be seen at, for example, a finer 2-mm³ voxel level[61]. Therefore, it would be necessary to examine the current observations with finer spatial resolution and with a larger number of regions of interest.

Our work has identified atypical brain dynamics of high-functioning adults with ASD by applying energy-landscape analyses to resting-state fMRI data. This data-driven approach has revealed that brains of ASD individuals are less dynamic than those of neurotypical controls, and such overly stable neural dynamics underlie core symptoms of ASD. In addition, we have shown that—in contrast to neurotypical individuals—such aberrantly stable brain dynamics do not impair but rather support general cognitive ability in ASD. Moreover, we have found that specific across-network functional coordination underpins such atypically stable brain dynamics in autism. These findings indicate the possibility that atypical brain dynamics might be a key biological endophenotype of ASD, and show that investigating brain dynamics can potentially make substantial contributions to our understanding of neural mechanisms underlying various typical/atypical behaviours and cognitive abilities.

## Methods

**Participants.** The current study used resting-state fMRI and anatomical MRI data shared in ABIDE[29]. To exclude effects of multiple recording sites, we analysed neuroimaging data collected in a single site (here University of Utah). This recording site was chosen because their data had the largest number of high-functioning adult males with ASD.

We selected participants based on their age ($18 \leq \text{age} \leq 40$), sex (male), handedness (right-handed), IQ ($80 \leq \text{full/verbal/performance IQ} \leq 140$), and head motion during echo planar imaging (EPI) recording (mean head motion $\leq 3$ mm).

To reduce confounding effects of age on IQ, we set the upper limit of the age at 40. Male participants were chosen because the number of female participants was substantially smaller than that of male participants in the data set, and it was practically impossible to balance the female/male proportion. Intelligence of the participants was quantified by the four subtests of the Wechsler Abbreviated Scale of Intelligence, and their handedness was evaluated by the Edinburgh Handedness Inventory. The mean head motion was measured in preprocessing procedures of EPI data (see section on 'Data Preprocessing' below).

As a result of this selection, this study used data obtained from 24 high-functioning adult males with ASD and 26 age-/sex-/IQ-matched neurotypical individuals (Table 1).

The ASD participants were diagnosed by a clinical autism expert in accordance with ADOS[32] and Diagnostic and Statistical Manual of Mental Disorders, Fourth Edition, Text Revision (DSM-IV-TR). In this data set, 16 of the 26 TD individuals were given their ADOS scores.

This data collection was approved by the local ethic committee of the recording site (University of Utah IRB), and all participants signed a written consent.

**Neuroimaging data.** The MRI data were collected in a 3T scanner (Magnetom Trio, Siemens). Resting-state fMRI data were recorded with an EPI sequence (TR 2 s, TE 28 ms, 40 slices, interleaved, flip angle 90°, spatial resolution 3.4 × 3.4 × 3.0 mm), whereas anatomical images were taken with T1-weighted sequence (repetition time (TR) 2.3 s, echo time (TE) 2.91 ms, flip angle 9°, spatial resolution 1.0 × 1.0 × 1.2 mm). For each participant, the EPI data were recorded in ∼8 min, while participants were instructed to relax with their eyes open.

**Data preprocessing.** These EPI data were preprocessed with SPM12 (www.fil.u-cl.ac.uk/spm) in essentially the same manner as in our previous study applying the energy-landscape analysis to resting-state fMRI data[26,27]: after discarding the first five images, the data underwent realignment, unwarping, slice timing correction, normalization to the standard template (ICBM 152) and spatial smoothing (Gaussian kernel with 8 mm of full-width at half maximum). We then removed effects of head motion, white matter signals, cerebrospinal fluid signals and global signal. Finally, we performed band-pass temporal filtering (0.01–0.1 Hz).

We then extracted a time series of fMRI signals from each of 214 regions of interest (ROIs) that were selected from the 264 ROIs listed in the previous studies[30]. The other 50 ROIs were not adopted here because they were labelled 'Uncertain' or 'Subcortical' and did not constitute specific cortical networks. The 214 cortical ROIs were defined as 4-mm spheres around the centre coordinates that were determined in the previous studies[30]. On the basis of these prior studies, we then classified the ROIs into seven functionally different brain networks (Fig. 1a). For each participant, we calculated seven time series data that represent average network activity of the seven brain networks (Fig. 1b). For the following energy-landscape analysis, we then binarized the network activity data in each participant using the average brain activity value[26,27,31,62]. Technically, in each participant, we first calculated the average brain activity for each network and binarized the original brain activity using the average activity as a threshold. Finally, we concatenated them across participants. As a result, for each of the TD and ASD group, we obtained seven binary time series data representing seven network activity (+1 for active, −1 for inactive). This binarization process enabled us to balance the number of active state and that of inactive state for each network, which could reduce the risk of overfitting in the following analysis[63].

The dorsal/ventral attention networks and cingulo-opercular network were merged into the ATN because (i) the current data size is not enough to perform energy-landscape analysis with nine factors and (ii) these three networks are considered to be responsible for the similar attention-related cognitive activity[30]. In fact, in the binary form, brain activity patterns of these three networks were significantly similar to each other ($\geq 71\%$ in TD, $\geq 73\%$ in ASD, $P \leq 10^{-5}$ in one-sample t-tests across participants; Supplementary Fig. 13a,b), whereas those of the other networks did not show such high similarity ($< 58\%$). In addition, even after we divided the ATN into cingulo-opercular network and the other two systems (dorsal/ventral attention networks), we observed qualitatively the same energy landscapes with the same six local minima (Supplementary Fig. 13c,d). These results are considered to justify our merging the three networks into one system.

These network activity patterns sufficiently represented the activity of cortical brain regions that the networks cover. In all the seven networks, the similarity between the binary network activity and neural activity of its constituent ROIs was significantly high across participant (mean similarity $\geq 69\%$, $P < 10^{-4}$ in a binomial test; Supplementary Fig. 14a). We can see such high similarity even at a voxel level across participant (mean similarity $\geq 66\%$, $P < 10^{-3}$; Supplementary Fig. 14b).

The similarity between brain activity $X_i$ and $X_j$ was calculated as $\left(N_T - \left\| X_i - X_j^2 \right\| \right) / N_T$, where $X_i$ were vectors representing time series of the binarized neural activities (1 for active, 0 for inactive) of brain network $i$ (or ROI$_i$) and $N_T$ denoted the length of $X_i$.

**Model fitting.** As a preparation for the following energy-landscape analysis, we fitted a pairwise MEM to the seven binary time series data in essentially the same manner as in previous studies[26,27,31,62].

First, every network activity pattern at time $t$ was described such as $V^t = [\sigma_1^t, \sigma_2^t, \ldots, \sigma_N^t]$, where $\sigma_i^t$ represents a binary activity of network $i$ at time $t$ (that is, $\sigma_i^t = +1$ or $-1$) and $N$ is the number of the networks (that is, $N = 7$, here, Fig. 1a). According to the principle of maximum entropy[64], when the mean network activity $\langle \sigma_i \rangle$ and the mean pairwise interaction $\langle \sigma_i \sigma_j \rangle$ are constrained by the empirical data, the appearance probability $P(V_k)$ of a network activity pattern $V_k$ should obey Boltzmann distribution[64], because such a distribution maximizes the information entropy[64]. That is, the appearance probability should be stated as

$$P(V_k) = e^{E(V_k)} / \Sigma_{i=1}^{2^N} e^{-E(V_k)}, \tag{1}$$

where

$$E(V_k) = -\Sigma_{i=1}^{N} h_i \sigma_i(V_k) - \frac{1}{2} \Sigma_{i=1}^{N} \Sigma_{j=1, j \neq i}^{N} J_{ij} \sigma_i(V_k) \sigma_j(V_k). \tag{2}$$

Here $\sigma_i(V_k)$ is the binarized activity of network $i$ in the activity pattern $V_k$, whereas $h_i$ represents basal activity of network $i$ and $J_{ij}$ indicates a pairwise interaction between networks $i$ and $j$.

Using this $P(V_k)$, we could calculate the model-based mean network activity $\langle \sigma_i \rangle_m = \Sigma_{\ell=1}^{2^N} \sigma_i(V_\ell) P(V_\ell)$ and model-based mean pairwise interaction

$\langle\sigma_i\sigma_j\rangle_m = \Sigma_{\ell=1}^{2^N}\sigma_i(V_\ell)\sigma_j(V_\ell)P(V_\ell)$. Therefore, we adjusted $h_i$ and $J_{ij}$ until these $\langle\sigma_i\rangle_m$ and $\langle\sigma_i\sigma_j\rangle_m$ were approximately equal to the empirically obtained $\langle\sigma_i\rangle$ and $\langle\sigma_i\sigma_j\rangle$. These adjustments of $h_i$ and $J_{ij}$ were conducted using a gradient ascent algorithm for each of the TD and ASD groups.

Accuracy of this model fitting was evaluated by (i) calculating a Pearson correlation coefficient between the model-based appearance probability and empirically obtained appearance probability, and (ii) estimating a proportion of Kullback–Leibler divergence in this 2nd-order model ($D_2$) against that in the 1st-order model ($D_1$) as follows[26,31]: $(D_1 - D_2)/D_1$ (Fig. 1f).

## Identification of local minima.

We built an energy landscape and searched for dominant brain states as in our previous studies[26,27]; we summarize the method below.

The energy landscape was first defined as a network of brain activity patterns $V_k$ ($k = 1, 2, \ldots, 2^N$) with their energy $E(V_k)$, in which two activity patterns were regarded as adjacent if and only if they took the opposite binary activity at a single brain network. We then searched for local energy minima, whose energy values were smaller than those of all the $N$ adjacent patterns.

To examine hierarchal structures between the detected local minima, we then constructed disconnectivity graphs[65] as follows[26,27]. (i) We prepared a so-called hypercube graph, in which each node representing a brain activity pattern was adjacent to the $N$ neighbouring nodes. (ii) We set a threshold energy level, $E_{th}$, at the largest energy value among the $2^N$ nodes. (iii) We removed the nodes whose energy values were $\geq E_{th}$. (iv) We examined whether each pair of local minima was connected by a path in the reduced network. (v) We repeated steps (iii) and (iv) after moving $E_{th}$ down to the next largest energy value. We ended up with a reduced network in which each local min was isolated. (vi) On the basis of the obtained results, we built a hierarchical tree whose leaves (that is, terminal nodes down in the tree) represented the local minima and internal nodes indicated the branching points of different local minima.

## Estimation of the sizes of dominant brain states.

To calculate how dominant each local min was, we then calculated basin sizes of the detected local minima (Fig. 1c). We first selected a starting node $i$ from the $2^N$ nodes. If any of its neighbour nodes has a smaller value of energy than node $i$, we moved to the neighbour node with the smallest energy value. Otherwise, we did not move, which indicated that node $i$ was a local min. We repeated this procedure until we arrived at a local min, and the starting node $i$ was defined as an element of the basin of the local min that was finally reached. We repeated this process for all the $2^N$ nodes. The basin of a local min was determined as a set of the brain activity patterns belonging to the basin, and the basin size was defined as the fraction of the number of the brain activity patterns belonging to the basin.

In this study, based on the hierarchal structures of the local minima, six local minima were summarized to two major and two minor brain states. The sizes of these dominant brain states were defined as the summation of the basin size of the constituting local minima (for example, the size of the major state #1 = (the basin size of local min $A$) + (the basin size of local min $B$); Fig. 2a). The distribution of the brain state size was compared between the TD and ASD groups by conducting a $\chi^2$-test and post hoc residual tests with Bonferroni correction.

On the basis of such definitions of the four brain states, we also estimated the appearance frequency of each brain state using the empirical data. Technically, for each participant, we counted how many times brain activity patterns belong to each dominant brain state were observed in the preprocessed fMRI data. The appearance frequency was compared between the TD and ASD individuals by two-sample $t$-tests with Bonferroni correction.

## Random-walk simulation of brain dynamics.

To characterize brain dynamics, we performed a random-walk simulation in the energy landscape in the same manner as in our previous study[26] (Fig. 1d). Technically, we numerically simulated movement of the brain activity patterns using a Markov chain Monte Carlo method with the Metropolis–Hastings algorithm[66,67]. In this method, any brain activity pattern $V_i$ could move only to a neighbouring pattern $V_j$ with probability $P_{ij} = \min\left[1, e^{E(V_i) - E(V_j)}\right]$. For each group, we repeated the random-walk $10^5$ steps with a randomly chosen initial pattern, and summarized the trajectory of activity patterns to a series of stays and transitions among the two major and two minor brain states. The first 100 steps were discarded to eradicate the effects of the initial condition. On the basis of this numerical simulation, we estimated transition frequency between the four dominant brain states and duration of staying in the major states.

As with the case of the estimation of the brain state size, we validated these simulation-based results by directly counting the empirical data. Note that this direct counting of the actual fMRI data was performed for each participant. Therefore, using the counting results, we performed partial correlation analyses and examined relationships between three indices of brain dynamics (that is, the duration of the major states, indirect transition frequency and intermediate-state frequency).

Differences in brain dynamics between the ASD and TD groups were evaluated initially in parametric tests (here, $t$-tests) but also confirmed with non-parametric approaches using permutation tests with $10^4$ random permutations. The number of

random-walk steps was set at $10^5$ because the two simulation-based indices for brain dynamics (that is, the duration of the major states and indirect transition frequency) showed sufficiently small fluctuation after $10^5$-step simulation (coefficient of variation < 0.005, Supplementary Fig. 15).

## Associations between brain dynamics and behaviours.

Next, we investigated associations between brain dynamics and behaviours (Fig. 1e). Technically, we calculated Pearson correlation coefficients between two behavioural indices (ADOS total scores and FIQ) and two brain dynamic indices sensitive to autism (the indirect transition frequency and duration of the major states). In estimation of a relationship between the indirect transition frequency and autistic traits of TD participants, the data of 16 of the 26 TD individuals were used because the data set did not contain ADOS scores for the other eight TD individuals.

## Functional coordination between brain networks.

Finally, we examined associations between across-network functional coordination and atypical brain dynamics, because, conceptually, neural phenomena occurring in short-time intervals (here, brain dynamics) could be constrained by long-term static functional structures (here, resting-state FC)[68,69].

Technically, we first calculated the strength of functional segregation between a network module (DMN, SMN and Auditory network) and another network module (FPN, SAN, ATN, Visual network) because the intermediate brain state was based on segregation between these two network modules (Fig. 6a). The strength of functional segregation was defined as the difference between the average of within-module FC and the average across-module FC (Fig. 6b). FC was calculated as a Pearson correlation coefficient between the average network activity, and the functional segregation strength was estimated based on the Fisher-transformed FC values (that is, Z-scores). Finally, we compared associations between this functional segregation strength and brain dynamics and behavioural scores. We performed the logically same analysis for the major brain states (Fig. 7).

Mathematically, if $FC_{ij}$ between the brain networks $i$ and $j$ is sufficiently close to $J_{ij}$ calculated by the pairwise MEM (Supplementary Fig. 1b), this FC-based index representing the functional segregation strength should be highly correlated with the flexibility of brain dynamics for the following reason. Now, in a given brain state, if the brain networks $i$ and $j$ belong to the same module, $\chi_i\chi_j$ is always 1 in the equation to calculate the energy value of the state (that is, equation 2). In contrast, if the brain networks $i$ and $j$ belong to the different modules, $\chi_i\chi_j$ is always $-1$. Thus, for example, during the intermediate state, increases in the within-module FCs ($J_{ij}$) and decrease in the across-module FCs ($J_{ij}$) should decrease the energy values of the intermediate state, increase the intermediate-state frequency and enhance the indirect transition frequency. That is, if $FC_{ij}$ is equal to $J_{ij}$, the magnitude of the functional segregation during the intermediate state should be positively correlated with the indirect transition frequency.

However, $FC_{ij}$ is theoretically not equal to $J_{ij}$, because FC (unlike $J_{ij}$) is based on Pearson correlation and does not consider effects of pairwise interactions. In fact, our previous study showed that FC is significantly different from $J_{ij}$ in terms of similarity to anatomical connections[32]. Thus, it is not mathematically trivial to examine the associations between the functional segregation and brain dynamics. Note that, in both TD and ASD groups, the functional segregation strength was not significantly correlated with global connectivity levels ($|r| \leq 0.16$, $P \geq 0.42$, Supplementary Table 4), which suggests that observations about functional segregation cannot be explained simply by global network-wide connectivity.

## Reproducibility tests using different data sets.

We tested the reproducibility of the current results using two other data sets in ABIDE[29]: data recorded in Indiana University (9 ASD and 12 TD individuals; TR 0.813 s; TE 28 ms; Flip angle 60° for fMRI data; Supplementary Table 2) and those collected in ETH Zürich (10 ASD and 15 TD individuals; TR 2 s, TE 25 ms, Flip angle 90° for fMRI data; Supplementary Table 3).

These two data sets were chosen because, except the data of the University of Utah, they had the largest or second largest MRI images of ASD adult males in the ABIDE database. Using these data, we first selected high-functioning ASD adult males and age-/sex-/IQ-matched TD individuals based on the same criteria as those applied to the data of the University of Utah. We then performed the same analyses using these independent data, and examined the robustness of the current observations.

## Reproducibility tests using different brain parcellation methods.

We also tested the robustness of the current observations using two different definitions of brain networks because choices of brain parcellation methods could affect results of some calculations about large-scale brain architecture[70].

In one of the new brain parcellation methods, which was a finer version of the original brain network definition, we divided the DMN into two subnetworks according to a previous study[40]. Technically, we classified the DMN ROIs into two groups: ROIs whose activities were mainly correlated with those of ventromedial prefrontal cortex (vmPFC, [2, 54, –3] in Talairach coordinates) and ROIs whose activities were mainly correlated with those of posterior cingulate cortex (PCC, [–2, –51, 27] in Talairach coordinates). The vmPFC and PCC were chosen because the

previous study showed their distinct roles in whole-brain network coordination[40]. The anatomical coordinates of the regions were determined based on the study.

For example, we categorized an ROI in DMN as follows. (i) For each participant, we calculated two Pearson correlation coefficients between the brain activities of the $ROI_i$, vmPFC and PCC (that is, an $ROI_i$–vmPFC correlation and an $ROI_i$–vmPFC correlation). (ii) After applying Fisher's Z-transformation to the correlation coefficients and averaging the Z-scores across all the participants, we compared the average $ROI_i$–vmPFC correlation with the average $ROI_i$–PCC correlation. (iii) If the $ROI_i$–vmPFC correlation was larger than the $ROI_i$–PCC correlation, the $ROI_i$ was labelled as a region of vmPFC-DMN. Otherwise, the $ROI_i$ was categorized as a region of PCC-DMN. (iv) We repeated this calculation for all the 59 ROIs of the DMN.

As a result of this procedure, we obtained a vmPFC-DMN and a PCC-DMN. The other six networks were the same as those in the original analysis. In total, we determined eight brain networks (vmPFC-DMN, PCC-DMN, FPN, SAN, ATN, SMN, Auditory and Visual), and repeated the same energy-landscape analysis for the eight brain systems.

The other brain parcellation was adopted from a different previous study[41]. In the study, the cortical area was divided into seven brain systems that were not exactly the same as those used in the original analyses (Supplementary Fig. 11a). Technically, we estimated the average neural activity for each brain system using a '7 network tight mask' (surfer.nmr.mgh.harvard.edu/fswiki/CorticalParcellation_Yeo2011), and repeated the entire analysis. This previous study also proposed a 17-network brain parcellation method, but we did not choose it because the size of the current data set was too small to accurately perform energy-landscape analysis for such 17 networks.

These robustness tests against differences in brain parcellation methods were conducted using the data set collected at the University of Utah.

**Diagnosis prediction.** Using the independent data sets recorded in Indiana University and ETH Zürich, we examined whether differences in brain dynamics can distinguish ASD individuals from TD individuals (Supplementary Fig. 12a). First, for each participant in the two independent data sets, we performed the energy-landscape analysis and calculated the intermediate-state frequency and the indirect transition frequency. We focused on these two indices for brain dynamics because they were clearly different between the TD and ASD groups (Fig. 3, f).

In a univariate diagnostic prediction, we set the cutoff values based on the original observations as follows: ((the minimum value in the TD group) + (the maximum value in the ASD group)) × 0.5.

In a multivariate prediction, we determined the cutoff line by fitting a linear support vector machine to the original data sets (Fig. 9c). This fitting was performed using LIBSVM package (www.csie.ntu.edu.tw/~cjlin/libsvm/).

We then applied these cutoff values and cutoff line to the two indices for brain dynamics observed in the independent data sets, and calculated sensitivity and specificity of the prediction.

**Data availability.** All the data used in this study are available in ABIDE[29] (the data sets collected in the University of Utah School of Medicine, Indiana University and ETH Zürich).

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

## Acknowledgements

This work was supported by the Wellcome Trust (G.R.) and the European Commission (T.W.).

## Author contributions

T.W. and G.R. designed the study. T.W. analysed the data. T.W. and G.R. wrote the manuscript.

## Additional information

**Competing interests:** The authors declare no competing financial interests.

