## [Peer Review File · Nature Communications]

Reviewers' comments:

Reviewer #1 (Remarks to the Author):

The authors present results of an energy-landscape analysis applied to resting state fMRI data collected from adults with autism and neurotypical adults. They find that the brains of individuals with autism show fewer transitions, and that frequency of transitions predicted severity of symptoms. They also report relationships with strength of functional segregation between brain networks and symptoms. This is an interesting approach applied to an important clinical question. Some suggestions are below.

I appreciate the approach to focus on data collected at one site to avoid issues related to cross-site comparison. However, the study would be greatly increased by demonstrating replicability of the current results over several additional sites containing comparable adult datasets.

The main finding is that minor states appeared less frequently in the autism group than in the control group. Can the authors comment on the potential impact between-subject heterogeneity and whether this may affect the autism and control groups differently?

The finding of increased state duration in autism seems more likely to be related to symptoms of rigidity, rather than social communication deficits. Did the authors examine relationships between the dynamic metrics and scores on subscales of the ADOS to test for this?

Did the authors test for age-related effects?

Reviewer #2 (Remarks to the Author):

Brain network dynamics in high-functioning autism

December 12, 2016

Comments

In this submission the authors leverage an energy landscape and maximum entropy approach to model brain-wide dynamics in a cohort of high-functioning autistic adults and a control group. Using this approach they show that transitions to a particular brain state (a vector of binarized activity profile) occur less frequently as a function of autism symptom severity. They go on to show other relationships between measures derived from the dynamical model and behavior. Finally, they show that the extent to which brain networks are segregated also differ between the two groups.

Overall, the submission is quite stanard in that the authors use an established set of techniques (energy landscape analysis, which they have applied before [4, 5]) to compare a control cohort with a clinical cohort (in this case, patients with autism). It goes a bit beyond the run-of-the-mill article in that its three principal findings (infrequent transitions to intermediate state, relation of cognitive performance with measures from the model, and network segregation differences) each could be crafted into a separate paper. Nonetheless, due to the overlap with the authors' past work, the novelty of the present submission is lessened.

I am not an expert in autism and social cognitive disorders, so it is difficult for me to judge how the findings reported in this submission will influence current autism research. From a methodological standpoint, however, there are a number of shortcomings that should temper enthusiasm over the findings and should be discussed in greater detail by the authors and/or addressed with supplementary analyses.

One of the greatest shortcomings, in my opinion, is the limitations imposed on the analysis by the maximum entropy model. Fitting the model becomes computationally challenging when the number of nodes/brain regions is greater a handful (which is why the authors group brain regions into seven systems rather than analyzing all nodes). While there is ongoing debate over the "correct" number of brain regions/nodes/systems into which the brain should be divided (there is likely a degeneracy of solutions for this problem), fixing that number at seven seems like a far too few as it almost certainly overlooks nuanced functional and anatomical connectivity profiles of the systems' sub-regions. I expand on this in the next section.

1. (p. 4) The accuracy of model fit is based on binarized, system-averaged time series. It is unclear, however, whether those system time series accurately represent the time series of each system's constituent voxels/brain regions. The authors could demonstrate a good match by calculating some measure of voxel-wise homogeneity (e.g. calculate the fraction of variance explained in voxel time series by the system time series).
2. (p. 5) The compare controls-ASD using t-tests. It would be good to see similar results with non-parametric tests; e.g. compute effect size and compare to random re-assignments of individuals to each group.
3. (p. 5) Why 10^5 steps and not some other number of steps? Does the frequency with which states are visited stabilize by that point?
4. (p. 9) The authors assert that transition frequency is associated with the strength of module segregation (as measured using Pearson correlation). My sense is that this finding might be a mathematical inevitability stemming from the interdependence of dynamics and FC. Variation in FC across time is constrained

by the organization of “static” (long-time averaged) FC [3, 2, 1]. The correlation magnitude of two times series over short intervals is constrained by the correlation over long intervals, with strong constraints for high-amplitude correlations and liberal constraints for low-amplitude correlations.

Segregated modules imply strong within-module correlations and equally strong between-module anti-correlations. Thus, a possible interpretation is that both the increased appearance of an intermediate state (due to liberal dynamical constraints) and the weak segregation are manifestations of the same phenomenon.

5. (p. 9) Change “weaken” to “weaker”.
6. (p. 14) The abbreviation should be “WASI”.
7. (p. 14) What is the rationale for selecting data from just the Utah site? The authors could leverage the other sites for reproducibility analysis, even if the data quality/number of subjects/controls are less than ideal.
8. (p. 15) Why 214 ROIs? The paper the authors reference includes 264.
9. (p. 15) I would expect that cognitive control and attentional networks play distinct roles in social disorders like ASD. I recognize the convenience of combining them for computational reasons, but is this actually justified? It would be good if the authors could either demonstrate that these three systems independently follow similar time series (thus justifying the merger) or reproduce their analyses by adding either one (split ATN but retain merger of attention networks) or two (split ATN into three components) new nodes.
10. (p. 15) The authors note that activity was binarized using the “average brain activity value”. They reference several papers but provide no further explanation as to what this entails. It would be good to be didactic in this explanation. I am also curious, then, about the robustness of their findings with respect to alternative thresholds (as the technique for selecting the threshold they reference is one of many and, in and of itself, is not privileged in any real way).
11. (p. 19) Another possible concern is that the segregation of systems (which the authors define as the different of within - between module connectivity) could be influenced by global connectivity levels. It would be good to show that the segregation measure is not significantly correlated across subjects with any of (*A*) sum of absolute value of all connections, (*B*) sum of all connections, (*C*), sum of only positive connections, or (*D*) sum only negative connections. If any of these correlations are strong, it suggests that the segregation measure may, in fact, be driven by network-wide connectivity differences rather than connectivity differences specific to the systems in question.

I hope that the authors find these comments useful.

References

- [1] Richard F Betzel, Makoto Fukushima, Ye He, Xi-Nian Zuo, and Olaf Sporns. Dynamic fluctuations coincide with periods of high and low modularity in resting-state functional brain networks. *NeuroImage*, 127:287–297, 2016.
- [2] Makoto Fukushima, Richard F Betzel, Ye He, Xi-Nian Zuo, and Olaf Sporns. Characterizing spatial patterns and flow dynamics in functional connectivity states and their changes across the human lifespan. *arXiv preprint arXiv:1511.06427*, 2015.
- [3] William H Thompson and Peter Fransson. The mean–variance relationship reveals two possible strategies for dynamic brain connectivity analysis in fmri. *Frontiers in human neuroscience*, 9, 2015.

- [4] Takamitsu Watanabe, Satoshi Hirose, Hiroyuki Wada, Yoshio Imai, Toru Machida, Ichiro Shirouzu, Seiki Konishi, Yasushi Miyashita, and Naoki Masuda. A pairwise maximum entropy model accurately describes resting-state human brain networks. *Nature communications*, 4:1370, 2013.
- [5] Takamitsu Watanabe, Naoki Masuda, Fukuda Megumi, Ryota Kanai, and Geraint Rees. Energy landscape and dynamics of brain activity during human bistable perception. *Nature communications*, 5, 2014.

Reviewer #3 (Remarks to the Author):

This manuscript uses an energy-landscape of resting state fMRI data to compare brain network dynamics in individuals with ASD and TD individuals. Very little research has probed brain network dynamics in autism, but based on theoretical and empirical work it is likely disrupted in autism. Therefore, this is an innovative and important addition to the literature. The authors found that in ASD, brain states are more stable and show fewer transitions of intermediate states than in TD individuals, and that these atypical dynamics are related to ASD symptomatology and IQ.

Following are suggestions to make this manuscript more suitable for publication:

This is a minor point, but I believe it's more standard to refer to people who have a diagnosis of autism as individuals with autism or ASD, and their brains as the brains of individuals with autism/ASD (or phrases similar to that), as opposed to "autistic" individuals or brains.

A relevant article that you did not cite is: Uddin, Supekar, Lynch, Cheng, Odriozola, Barth, Phillips, Feinstein, Abrams & Menon (2015). Brain state differentiation and behavioral inflexibility in autism. *Cerebral Cortex*, 25, 4740-7. It is one of the few articles to date that has measured temporal changes in functional connectivity in autism, and supports your findings that brain states are too stable in autism.

Did you look at data for any of the other ABIDE sites? Given that you chose to only look at a single site because of potential across-site differences and confounds, it would be useful to be able to demonstrate that these results are not a quirk of the single site you chose. Relating Utah results to results from one of the other sites would be informative.

In Table 1, why not include p-values comparing ADOS scores across TD and ASD groups? It would be a nice way to emphasize the expected difference in scores.

On page 4, line 113, you write "Coincidentally, we identified the same six locally stable brain activity patterns in the TD and ASD individuals..." Why do you think this is a coincidence? You write that in the Figure 2 caption as well. This finding is consistent with the work from Vince Calhoun's lab demonstrating that different populations (i.e., schizophrenia versus healthy control subjects) have the same brain states, but different dwell time and transition patterns across brain states. Most likely, it is meaningful that individuals with autism have the same brain activity patterns as healthy individual, but different transition patterns and dynamics. This should be written about in the discussion as well.

Overall, your figures are very busy and therefore hard to follow. Perhaps separating them into more figures, spreading the different plots out more, shortening the axis titles, and adding legends so you do not have to have words written over the plots, would help.

Page 7, line 199: "...whereas such difference was seen in duration of Major states (P >

0.56). " I do not understand what this phrase means, please elaborate in the text. In that same paragraph, what is the ADOS score you are correlating? Overall score? One of the subscales? Please specify.

The last sentence of that section: "These observations indicate that atypically unstable Intermediate state in autistic brains reduces their Indirect transition, which increases the severity of autistic symptoms (Figs. 4c)." is a bit strong, given that you do not know if the decreased indirect transition is what causes increased severity of ASD symptomatology. From this analysis you know that decreased indirect transitions are related to increased ASD symptomatology. Whether it is a direct cause or a hierarchical relationship caused by an unmeasured variable you do not know from this analysis. The same goes for the other results of similar analyses that you describe throughout the text.

In Figure 4a, if you remove and/or deweight the outlier with the low ADOS score and the highest indirect transition frequency, does the reported relationship still hold?

Last line of page 8 "...and a module with the other four networks..." it would be helpful for the reader to list those four networks out. Same goes for the 5 networks mentioned on lines 279 and 296 on page 10.

In Figures 6 and 7 (and the corresponding results section), did you also relate major state functional segregation to IQ in TD, and intermediate state functional segregation to ASD? It would be interesting to report those results.

The discussion should be fleshed out a bit :

- 1) What do you make of the different relationship between FIQ and brain states in TD and ASD? This is an interesting result and should be written about in the discussion.
- 2) If you were to look at connectivity patterns across the 7 systems that describe each of these states, how do these patterns relate to what we know about network organization in ASD based on previous literature?
- 3) Why is it that increased segregation during intermediate states and decreased segregation during major states is more typical/relevant to healthy cognition and brain dynamics?

In the Data Preprocessing section of the methods (page 15), you can quantify if the timeseries of the DAN, VAN, and CO networks are similar enough to justify averaging together. Please report that.

The authors' response to some of the concerns raised by the Reviewers that were highlighted in the original decision letter.

We have addressed all the comments of the editor and reviewers. Page numbers in our responses refer to those in the revised manuscript unless otherwise mentioned. The modified or added text is in **blue** in the revised manuscript.

[1] Reproducibility.

First, we confirmed the reproducibility of the original observations using two independent datasets, which were collected in the Indiana University (9 ASD and 12 TD individuals) and ETH Zürich (10 ASD and 15 TD individuals). We clarified this reproducibility by adding four new supplementary figures (Supplementary Figs. 6-9) and two tables (Supplementary Tables 2 and 3) with the following descriptions:

Results section (lines 9-21 in p. 11)

Reproducibility tests

We confirmed the reproducibility of these observations using two independent datasets collected in Indiana University (Supplementary Table 2) and ETH Zürich (Supplementary Table 3).

Analysis of both datasets yielded qualitatively the same hierarchal structures of the energy landscapes, consisting of the same six brain states (Supplementary Figs. 6b and 8b). In addition, significant differences in the major/intermediate state frequency between TD and ASD groups were also reproduced (Supplementary Figs. 6c and 8c). Moreover, we could identify the atypically lower indirect transition frequency and aberrantly longer duration of the major states in the ASD groups (Supplementary Figs. 6d-g, and 8d-g). Finally, we confirmed that the correlations between brain dynamics and behavioural indices were reproduced (Supplementary Figs. 6h-j and 8h-j), and the associations between these brain dynamics and across-network functional coordination were also replicated (Supplementary Figs. 7 and 9).

Methods section (lines 12-24 in p. 23)

Reproducibility test

We tested the reproducibility of the current results using two other datasets in ABIDE²⁹: data recorded in Indiana University (9 ASD and 12 TD individuals; TR 0.813s, TE 28ms, Flip angle 60° for fMRI data; Supplementary Table 2) and those collected in ETH Zürich (10 ASD and 15 TD individuals; TR 2s, TE 25ms, Flip angle 90° for fMRI data; Supplementary Table 3). These two datasets were chosen because except the data of the University of Utah, they had the largest or second largest MRI images of ASD adult males in the ABIDE database.

We first selected high-functioning ASD adult males and age-/sex-/IQ-matched TD individuals based on the same criteria as those applied to the data of the University of Utah. We then applied the same analyses to these data, and examined the robustness of the current observations.

[2] Diagnostic prediction

Second, according to the editor's suggestion, we examined whether the original findings can be used to distinguish ASD individuals from TD individuals. To this end, we defined thresholds for the classification using the original dataset (University of Utah dataset), and tested them using the two independent datasets (Indiana University and ETH Zürich datasets).

We found that the accuracy of the diagnostic prediction based on brain dynamics (Sensitivity = 89%, Specificity = 93% in a bi-variate analysis) was comparable or higher to those of previous diagnostic systems based on resting-state fMRI datasets (e.g., Yahata et al., 2016; Uddin et al., 2013; Anderson et al., 2011).

To clarify this point, we added a new figure (Fig. 9) with the following description.

Results section (line 23 in p. 11 – line 7 in p. 12)

Prediction of ASD diagnosis

Using the same independent datasets, we examined whether such differences in brain dynamics can potentially be used to predict the diagnosis of autism (Fig. 9 and Supplementary Fig. 10). We first defined a threshold for the diagnosis using the original dataset (University of Utah data), and then evaluated the performance of this diagnostic approach by applying it to the independent datasets (Indiana University and ETH Zürich data). For the test data, the energy-landscape analysis was performed not at a group but individual level, and therefore, the results were not exactly the same as those obtained in the above reproducibility test.

The intermediate state frequency could predict the ASD diagnosis with relatively high accuracy (sensitivity = 84%, specificity = 85%; Fig. 9a). In contrast, the indirect transition frequency did not realise such accurate prediction (sensitivity = 68%, specificity = 74%; Fig. 9b). We could improve the diagnosis accuracy by combining the two indices in a multi-variate pattern analysis method (sensitivity = 89%, specificity = 93%; Figs. 9c and 9d). Notably, this classification accuracy is comparable to or higher than previous diagnostic approaches based on resting-state fMRI data⁴¹⁻⁴³.

Methods section (line 25 in p. 23 – line 10 in p. 24)

Diagnosis prediction

Using the same independent datasets, we examined whether differences in brain dynamics can distinguish ASD individuals from TD individuals (Supplementary Fig. 10a). First, for each participant in the two independent datasets, we performed the energy-landscape analysis and calculated the intermediate state frequency and the indirect transition frequency. We focused on these two indices for brain dynamics because they were clearly different between the TD and ASD groups (Figs. 3d and 3f)

In a univariate diagnostic prediction, we set the cut-off values based on the original observations as follows: $[(\text{min value in the TD group}) - (\text{max value in the ASD group})] \times 0.5$.

In a multivariate prediction, we determined the cut-off line by fitting a linear support vector machine to the original datasets (Fig. 9c). This fitting was performed using LIBSVM package (www.csie.ntu.edu.tw/~cjlin/libsvm/).

We then applied these cut-off values and cut-off line to the two indices for brain dynamics observed in the independent datasets, and calculated sensitivity and specificity of the prediction.

We have also illustrated the findings in a new figure (Fig. 9)

References

38. Yahata, N. *et al.* A small number of abnormal brain connections predicts adult autism spectrum disorder. *Nat Commun* **7**, 11254 (2016).
39. Uddin, L. Q. *et al.* Salience network-based classification and prediction of symptom severity in children with autism. *JAMA Psychiatry* **70**, 869–879 (2013).
40. Anderson, J. S. *et al.* Functional connectivity magnetic resonance imaging classification of autism. *Brain* **134**, 3742–3754 (2011).

The authors' point-by-point response to the specific concerns raised by each Reviewer.

Responses to the reviewer #1's comments

We appreciate the reviewer's useful suggestions. We have addressed all the comments of the reviewer. Page numbers in our responses refer to those in the revised manuscript unless otherwise mentioned. The modified or added text is in **blue** in the revised manuscript.

Concern 1

"I appreciate the approach to focus on data collected at one site to avoid issues related to cross-site comparison. However, the study would be greatly increased by demonstrating replicability of the current results over several additional sites containing comparable adult datasets."

We agree with the importance of the reproducibility test. Therefore, according to the reviewer's suggestion, we examined and confirmed that all of our original results could be reproduced in two independent neuroimaging datasets, which were collected in the Indiana University (9 ASD and 12 TD individuals) and ETH Zürich (10 ASD and 15 TD individuals).

To clarify this reproducibility, we added four new supplementary figures (Supplementary Figs. 6-9) and two tables (Supplementary Tables 2 and 3) with the following descriptions:

Results section (lines 9-21 in p. 11)

Reproducibility tests

We confirmed the reproducibility of these observations using two independent datasets collected in Indiana University (Supplementary Table 2) and ETH Zürich (Supplementary Table 3).

Analysis of both datasets yielded qualitatively the same hierarchal structures of the energy landscapes, consisting of the same six brain states (Supplementary Figs. 6b and 8b). In addition, significant differences in the major/intermediate state frequency between TD and ASD groups were also reproduced (Supplementary Figs. 6c and 8c). Moreover, we could identify the atypically lower indirect transition frequency and aberrantly longer duration of the major states in the ASD groups (Supplementary Figs. 6d-g, and 8d-g). Finally, we confirmed that the correlations between brain dynamics and behavioural indices were reproduced (Supplementary Figs. 6h-j and 8h-j), and the associations between these brain dynamics and across-network functional coordination were also replicated (Supplementary Figs. 7 and 9).

Methods section (lines 12-24 in p. 23)

Reproducibility test

We tested the reproducibility of the current results using two other datasets in ABIDE²⁹: data recorded in Indiana University (9 ASD and 12 TD individuals; TR 0.813s, TE 28ms, Flip angle 60° for fMRI data; Supplementary Table 2) and those collected in ETH Zürich (10 ASD and 15 TD individuals; TR 2s, TE 25ms, Flip angle 90° for fMRI data; Supplementary Table 3). These two datasets were chosen because except the data of the

University of Utah, they had the largest or second largest MRI images of ASD adult males in the ABIDE database.

We first selected high-functioning ASD adult males and age-/sex-/IQ-matched TD individuals based on the same criteria as those applied to the data of the University of Utah. We then applied the same analyses to these data, and examined the robustness of the current observations.

Concern 2

“The main finding is that minor states appeared less frequently in the autism group than in the control group. Can the authors comment on the potential impact between-subject heterogeneity and whether this may affect the autism and control groups differently?”

We appreciate the reviewer’s thoughtful suggestion concerning the heterogeneity of the participants. The current study attempted to reduce such heterogeneity by restricting age/sex/IQ scores and excluding individuals with any neuropsychiatric comorbidity or medication. However, it is also the case that not all the potential confounding effects of heterogeneity were controlled. Therefore, some other factors, (perhaps genetic, but other factors are also possible), might affect the current observations.

In addition, as suggested in a recent review (Jack, A. & Pelphrey, 2017), the effects of such between-participant heterogeneity may potentially be larger in the analysis of ASD data than in that of TD data. In fact, the number of outliers was larger in the ASD data than in the controls (Figs. 2d and 2e).

To clarify this issue, we added the following paragraph into the Discussion section (lines 2-12 in p. 15).

Another limitation of our work concerns potential heterogeneity in the ASD group⁵⁶. We attempted to reduce confounding effects of such heterogeneity by focusing on high-functioning right-handed male adults without any psychiatric comorbidity including ADHD. However, this approach could not control all between-participant differences. For example, some recent studies have reported that neural responses of ASD individuals could be affected by their genetic patterns^{57,58}, and other studies have pointed out significant diversity in executive function even in ASD individuals⁵⁹. In addition, such heterogeneity could be larger in the ASD group than in TD cohort⁵⁶. Consistent with this, we observed more outliers in our ASD data than neurotypical control data: the ASD data had seven outliers ($>$ or $<$ Mean \pm 2SD) in the appearance frequencies of the major/minor brain states (Figs. 2d and 2e), whereas the TD had no outlier. Given such potential heterogeneity, the current observations will need to be further examined in more genetically and behaviourally homogeneous sub-groups of ASD.

References

56. Jack, A. & Pelphrey, K. A. Annual Research Review: Understudied populations within the autism spectrum - current trends and future directions in neuroimaging research. *J Child Psychol Psychiatry* (2017).
57. Rudie, J. D. *et al.* Autism-associated promoter variant in MET impacts functional and structural brain networks. *Neuron* **75**, 904–915 (2012).

58. Watanabe, T. *et al.* Oxytocin receptor gene variations predict neural and behavioral response to oxytocin in autism. *Soc Cogn Affect Neurosci* nsw150 (2016). doi:10.1093/scan/nsw150
59. Dajani, D. R., Llabre, M. M., Nebel, M. B., Mostofsky, S. H. & Uddin, L. Q. Heterogeneity of executive functions among comorbid neurodevelopmental disorders. *Sci. Rep.* **6**, 36566 (2016).

Concern 3

“The finding of increased state duration in autism seems more likely to be related to symptoms of rigidity, rather than social communication deficits. Did the authors examine relationships between the dynamic metrics and scores on subscales of the ADOS to test for this?”

We have sympathy with the thought that inflexible brain dynamics in autism may be linked with rigidity symptoms of ASD.

As the reviewer stated, we found a significant negative correlation between ADOS total score and the indirect transition frequency in autism (Fig. 4a). However, we could specify with confidence which core symptom of ASD was more related to these atypically stable brain dynamics.

In fact, the ASD individuals with higher ADOS RRB scores showed smaller frequency of the indirect transitions (Cohen’s $d \sim 0.6$). At the same time, inflexibility of brain dynamics in the ASD individuals was moderately correlated with the severity of ASD social symptoms (ADOS social + communication scores, $r \sim -0.3$; ADOS social, $r \sim 0.3$; ADOS communication, $r \sim -0.4$).

We clarified this issue by adding the following descriptions into the Results and Discussion sections with a new supplementary figure (Supplementary Fig. 4).

Results section (lines 24-25 in p. 7).

This brain-symptom association was not specific to either of the social or non-social core symptoms of autism (Supplementary Fig. 4).

Discussion section (lines 25- 31 in p. 14).

We also need to be careful not to conclude that the aberrantly stable brain dynamics of individuals with autism are related to every aspect of the disorder. Both the social and non-social core symptoms of ASD showed similar effect sizes for brain-symptom associations (Supplementary Fig. 4), but some previous studies have suggested the possibility that non-social symptoms are more relevant to such neurophysiological inflexibility^{10,25}. To identify which property of atypical brain dynamics is related to a specific core symptom of autism would require a combination of the current energy-landscape analysis with task-based neuroimaging data of ASD individuals.

We also added a new Supplementary Fig. 4

Associations between the Indirect transition frequency and sub-scale scores of ADOS.

The social and non-social ADOS scores commonly showed mild negative associations with the Indirect transition frequency in the ASD group, and there was no significant difference between the brain-symptom relationships (a and b). This tendency did not change even after we re-calculated the associations using all of the three datasets (c and d).

According to DSM-5, we quantified the social symptom of autism by merging ADOS social score and ADOS communication score. The correlations between the merged ADOS scores and brain dynamics were preserved when we calculated the correlations for each ADOS subscale separately (ADOS social $r \leq -0.32$, ADOS communication $r \leq -0.41$).

References

10. Dajani, D. R. & Uddin, L. Q. Demystifying cognitive flexibility: Implications for clinical and developmental neuroscience. *Trends in Neurosciences* **38**, 571–578 (2015).
25. Uddin, L. Q. *et al.* Brain State Differentiation and Behavioral Inflexibility in Autism. *Cereb. Cortex* **25**, 4740–4747 (2015).

Concern 4

“Did the authors test for age-related effects?”

We have now calculated the correlations between age and three indices for brain dynamics (i.e., the intermediate state frequency, the indirect transition frequency, and the duration of major states), but found no significant associations for any of these variables.

We have now reported these new analyses by adding the following descriptions into the Result section (lines 9-11 in p. 7) with a new supplementary table (Supplementary Table 1).

In both the TD and ASD groups, these three brain dynamic indices were not significantly correlated with the ages of the individuals ($|r| \leq 0.18$, $P \geq 0.37$; Supplementary Table 1).

Supplementary Table 1

Correlation coefficients (r) between brain dynamics and age

	Intermediate state freq.	Indirect transition freq.	Duration of Major states
TD	0.18	0.17	-0.12
ASD	-0.13	-0.16	-0.11

We thank the reviewer for their helpful comments which we believe have substantially improved our paper.

Responses to the comments of the editor and reviewers for “Brain network dynamics in high-functioning individuals with autism” by Watanabe and Rees (NCOMMS-16-26411).

Responses to reviewer #2’s comments

We appreciate the reviewer’s helpful suggestions and thorough review. We have replied to all the reviewer’s comments in the same order as used by the reviewer. The page numbers refer to those in the revised manuscript unless otherwise mentioned. The modified or added text is in **blue** in the revised manuscript.

Concern 1

“One of the greatest shortcomings, in my opinion, is the limitations imposed on the analysis by the maximum entropy model. Fitting the model becomes computationally challenging when the number of nodes/brain regions is greater a handful (which is why the authors group brain regions into seven systems rather than analyzing all nodes). While there is ongoing debate over the “correct” number of brain regions/nodes/systems into which the brain should be divided (there is likely a degeneracy of solutions for this problem), fixing that number at seven seems like a far too few as it almost certainly overlooks nuanced functional and anatomical connectivity profiles of the systems’ sub-regions. I expand on this in the next section.”

We appreciate the reviewer’s concern about this system-level way of approximating brain signals. However, please note that similar network-level signal approximations were adopted in parts of some previous neuroimaging studies (e.g., Allen et al., 2014; Baker et al., 2014; Chen et al., 2016; de Pasquale et al., 2012), and enabled those studies to report neurobiologically meaningful observations. Moreover, we found that in a binary form, such network-level activity was significantly similar to brain activity of the constituent ROIs (similarity $\geq 69\%$, $P < 10^{-4}$; Supplementary Fig. 12a) and voxel-wise fMRI signals (similarity $\geq 66\%$, $P < 10^{-3}$; Supplementary Fig. 12b). Thus, even though our approach may lose some information by being over-simplified, it is sufficient to derive reproducible (e.g. see our response to reviewer #1 major comment 1) findings that are associated with symptomatology and diagnostic category and therefore have face validity as neurobiological measures.

Nevertheless, as the reviewer stated, it is also the case that such seemingly crude approximation may lose detailed information about neural activity of far smaller brain regions. Therefore, to acknowledge this potential limitation, we have now added the following statements to the Discussion section and Methods section.

Discussion section (lines 14-20 in p.15)

Our analytic approach also had some methodological limitations. We classified cortical regions into the seven systems, and examined brain dynamics in terms of changes in the seven system-level brain activity (Figs. 1a and 1b). Although similar approximations have been partly adopted in other human neuroimaging studies and yield biologically meaningful observations^{2,3,5,7}, such system-level approximation of brain signals may lose detailed and nuanced information that should be seen at, for example, a finer 2-mm³ voxel level⁶⁰. Therefore, in future studies, it would be necessary to examine the current observations with finer spatial resolution and with a larger number of regions of interest.

References

- Allen, E. A. *et al.* Tracking whole-brain connectivity dynamics in the resting state. *Cereb. Cortex* **24**, 663–676 (2014).
- Baker, A. P. *et al.* Fast transient networks in spontaneous human brain activity. *eLife* **3**, e01867 (2014).
- Chen, T., Cai, W., Ryali, S., Supekar, K. & Menon, V. Distinct Global Brain Dynamics and Spatiotemporal Organization of the Salience Network. *PLoS Biol.* **14**, e1002469 (2016).
- de Pasquale, F. *et al.* A cortical core for dynamic integration of functional networks in the resting human brain. *Neuron* **74**, 753–764 (2012).
- Sporns, O., Tononi, G., & Kötter, R. The human connectome: A structural description of the human brain. *PLoS Comp Bio*, **1**, e42. (2005).

Concern 2

“1. (p. 4) The accuracy of model fit is based on binarized, system-averaged time series. It is unclear, however, whether those system time series accurately represent the time series of each system’s constituent voxels/brain regions. The authors could demonstrate a good match by calculating some measure of voxel-wise homogeneity (e.g. calculate the fraction of variance explained in voxel time series by the system time series).”

According to the reviewer’s suggestion, we compared the system-averaged brain activity with ROI-average/voxel-wise fMRI signals in a binary form, and found that the system-averaged brain activity was significantly similar to both types of the brain activity (mean similarity >66%, $P < 10^{-3}$ in binominal tests).

To clarify this issue, we added the following descriptions into the Methods section (lines 13- 17 in p. 18) with a new supplementary figure (Supplementary Fig. 12).

These network activity patterns sufficiently represented the activity of cortical brain regions that the networks cover. In all the seven networks, the similarity between the binary network activity and neural activity of its constituent ROIs was significantly high across participant (mean similarity $\geq 69\%$, $P < 10^{-4}$ in a binominal test; Supplementary Fig. 12a). We can see such high similarity even at a voxel level across participant (mean similarity $\geq 66\%$, $P < 10^{-3}$; Supplementary Fig. 12b).

Supplementary Fig. 12

Concern 3

“2. (p. 5) The compare controls-ASD using t-tests. It would be good to see similar results with non-parametric tests; e.g. compute effect size and compare to random re-assignments of individuals to each group.”

According to the reviewer’s suggestion, we calculated effect sizes of the differences between the ASD and TD groups, and re-evaluated the differences using permutation tests. This revealed effect sizes and P values very similar to the original ones seen from our use of parametric t -tests. We have now added these findings into the revised Results section as follows:

Results section

Lines 20-24 in p. 5

The two major brain states appeared more frequently in the ASD group than in the TD group ($t_{48} > 7.8$, $P_{\text{uncorrected}} < 10^{-9}$, $P_{\text{Bonferroni}} < 0.05$ in two-sample t -tests, $P = 0.0001$ in permutation tests, Cohen’s $d \geq 2.0$; Fig. 2d), whereas the two minor states showed significantly less appearance frequency in the ASD group ($t_{48} > 10.6$, $P_{\text{uncorrected}} < 10^{-13}$, $P_{\text{Bonferroni}} < 0.05$ in two-sample t -tests, $P = 0.0001$ in permutation tests, Cohen’s $d \geq 2.4$; Fig. 2e).

Lines 8-10 in p. 6

As with the minor brain states (Fig. 2e), the appearance frequency of this intermediate state was significantly smaller in the ASD group than in the controls ($t_{48} = 20.3$, $P_{\text{uncorrected}} < 10^{-5}$ in a two-sample t -test, $P = 0.0002$ in a permutation test, Cohen’s $d = 3.5$; Fig. 3d).

Lines 16-20 in p. 6

no significant difference was seen in the direct transition frequency ($t_{48} = 1.5$, $P = 0.13$ in a two-sample t -test, $P = 0.14$ in a permutation test, Cohen’s $d = 0.17$), whereas the indirect transition frequency was significantly lower in the ASD group ($t_{48} = 14.0$, $P_{\text{uncorrected}} < 10^{-5}$, $P_{\text{Bonferroni}} < 0.05$ in a two-sample t -test, $P = 0.0001$ in a permutation test, Cohen’s $d = 5.0$).

Lines 25-29 in p. 6

In this random-walk simulation, the ASD brains showed significantly longer duration of the major states than TD brains ($t_{8735} = 3.9$, $P < 10^{-4}$ in a two-sample t -test, $P = 0.0001$ in a permutation test, Cohen’s $d = 3.1$; Fig. 3g). This difference was reproduced in direct counting of the repetition length of the major states in the empirical data ($t_{48} = 3.6$, $P = 0.0008$ in a two-sample t -test, $P = 0.0013$ in a permutation test, Cohen’s $d = 1.0$; Fig. 3h).

Lines 22-25 in p. 9

However, the gap between the within- and across-module FCs was significantly smaller in the ASD group compared to the controls ($F_{1,96} = 272.2$, $P < 10^{-5}$ as an interaction in

the two-way factorial ANOVA; $t_{48} = 13.1$, $P < 10^{-5}$ in a post-hoc two-sample t -test, $P = 0.0001$ in a post-hoc permutation test, Cohen's $d = 3.0$; Fig. 6c), ...

Line 31 in p. 9 – line 2 in p. 10

Such a significant association between ADOS scores and the functional segregation strength was observed even in the TD data ($t_{16} = 2.5$, $P = 0.025$ in a two-sample t -test, $P = 0.0001$ in a post-hoc permutation test, Cohen's $d = 1.4$; Supplementary Fig. 5a).

Lines 20-24 in p. 10

Although significant functional segregation was seen in both the TD and ASD groups ($F_{1,96} = 72.3$, $P < 10^{-5}$ as a main effect of FC types in a two-way factorial ANOVA; Fig. 7c), its strength was significantly larger in the ASD individuals ($F_{1,96} = 5.6$, $P = 0.020$ as an interaction in a two-way factorial ANOVA; $t_{48} = 2.3$, $P = 0.02$ in a post-hoc two-sample t -test, $P = 0.03$ in a post-hoc permutation test, Cohen's $d = 0.63$; Fig. 7c).

Concern 4

“3. (p. 5) Why 10^5 steps and not some other number of steps? Does the frequency with which states are visited stabilize by that point?”

We set the number of random walk at 10^5 because the two indices representing brain dynamics showed sufficiently small fluctuation after such length of simulation. To confirm this, we repeated the random-walk simulation for each number of steps 1000 times, and calculated coefficients of variation (= standard deviation/mean) for each step.

We found that in both of the brain dynamics indices, the coefficients of variation reached a plateau after approximately 60000-step random walk.

We have now clarified this issue by adding the following description into the Methods section (lines 22-25 in p.21) and a new supplementary figure (Supplementary Fig. 13).

The number of random walk steps was set at 10^5 , because the two simulation-based indices for brain dynamics (i.e., the duration of the major states and indirect transition frequency) showed sufficiently small fluctuation after 10^5 -step simulation (coefficient of variation < 0.005 , Supplementary Fig. 13).

Supplementary Fig. 13

Concern 5

“4. (p. 9) The authors assert that transition frequency is associated with the strength of module segregation (as measured using Pearson correlation). My sense is that this finding might be a mathematical inevitability stemming from the interdependence of dynamics and FC. Variation in FC across time is constrained □ by the organization of “static” (long-time averaged) FC [3, 2, 1]. The correlation magnitude of two times series over short intervals is constrained by the correlation over long intervals, with strong constraints for high-amplitude correlations and liberal constraints for low-amplitude correlations. Segregated modules imply strong within-module correlations and equally strong between-module anti-correlations. Thus, a possible interpretation is that both the increased appearance of an intermediate state (due to liberal dynamical constraints) and the weak segregation are manifestations of the same phenomenon.”

We appreciate the reviewer’s important point. Conceptually, we agree with the reviewer’s argument that these short-term phenomena (i.e., brain dynamics) are constrained by long-term organisation (here, Pearson correlation-based FC). However, mathematically, it cannot be always the case because the current brain dynamics are calculated not based on conventional FC (i.e., Pearson correlation), but using a pairwise maximum entropy model (MEM).

If FC_{ij} between the brain networks i and j is exactly the same as J_{ij} calculated by the pairwise MEM, this FC-based functional segregation strength should be correlated with the flexibility of brain dynamics for the following reason.

Now, if the brain networks i and j belong to the same module, $\sigma_i \sigma_j$ is always 1 in the following equation to calculate energy values:

$$E(V_k) = -\sum_{i=1}^N h_i \sigma_i(V_k) - (1/2) \sum_{i=1}^N \sum_{j=1, j \neq i}^N J_{ij} \sigma_i(V_k) \sigma_j(V_k).$$

In contrast, if the brain networks i and j belong to the different modules, $\sigma_i \sigma_j$ is always -1 .

Thus, for example, during the Intermediate state, increases in the within-module FCs (J_{ij}) and decreases in the across-module FCs (J_{ij}) should decrease the energy values of the Intermediate state, increase the appearance frequency of the Intermediate state, and enhance the Indirect transition frequency. That is, if FC_{ij} is equal to J_{ij} , the magnitude of the functional segregation of the Intermediate state should be positively correlated with the Indirect transition frequency.

However, FC_{ij} is theoretically not equal to J_{ij} because FC (unlike J_{ij}) is based on Pearson correlation and does not consider effects of pairwise interactions. In fact, our previous study showed that FC is significantly different from J_{ij} in terms of similarity to anatomical connections (Watanabe et al., 2013).

Given such arguments, it is not mathematically trivial to examine the associations between functional segregation and brain dynamics.

To clarify this issue, we have now added the following statements into the Methods section.

Lines 7-10 in p. 22

Finally, we examined associations between across-network functional coordination and atypical brain dynamics, because conceptually, neural phenomena occurring in short-time intervals (here, brain dynamics) could be constrained by long-term static functional structures (here, functional connectivity)^{67,68}.

Line 22 in p. 22 – line 5 in p. 23

Mathematically, if FC_{ij} between the brain networks i and j is sufficiently close to J_{ij} calculated by the pairwise MEM (Supplementary Fig. 1b), this FC-based index representing the functional segregation strength should be highly correlated with the flexibility of brain dynamics for the following reason. Now, in a given brain state, if the brain networks i and j belong to the same module, $\sigma_i \sigma_j$ is always 1 in the equation to calculate the energy value of the state (i.e., eq. 2). In contrast, if the brain networks i and j belong to the different modules, $\sigma_i \sigma_j$ is always -1 . Thus, for example, during the intermediate state, increases in the within-module FCs (J_{ij}) and decrease in the across-module FCs (J_{ij}) should decrease the energy values of the intermediate state, increase the intermediate state frequency, and enhance the indirect transition frequency. That is, if FC_{ij} is equal to J_{ij} , the magnitude of the functional segregation during the intermediate state should be positively correlated with the indirect transition frequency.

However, FC_{ij} is theoretically not equal to J_{ij} , because FC (unlike J_{ij}) is based on Pearson correlation and does not consider effects of pairwise interactions. In fact, our previous study showed that FC is significantly different from J_{ij} in terms of similarity to anatomical connections³². Thus, it is not mathematically trivial to examine the associations between the functional segregation and brain dynamics.

References

32. Watanabe, T. *et al.* A pairwise maximum entropy model accurately describes resting-state human brain networks. *Nat Commun* **4**, 1370 (2013).
67. Thompson, W. H., & Fransson, P. The mean-variance relationship reveals two possible strategies for dynamic brain connectivity analysis in fMRI. *Front Hum Neurosci*, **9**, 398. (2015).
68. Betzel, R. F., Fukushima, M., He, Y., Zuo, X.-N., & Sporns, O. Dynamic fluctuations coincide with periods of high and low modularity in resting-state functional brain networks. *NeuroImage*, **127**, 287–297 (2016).

Concern 6

“5. (p. 9) Change “weaken” to “weaker”.”

Thank you - we have now changed the description (line 28 in p. 9).

Concern 7

“6. (p. 14) The abbreviation should be “WASI”.”

Thank you - we have now corrected this typo (line 29 in p. 16).

Concern 8

“7. (p. 14) What is the rationale for selecting data from just the Utah site? The authors could leverage the other sites for reproducibility analysis, even if the data quality/number of subjects/controls are less than ideal.”

We chose the original dataset because the data collected in the University of Utah contained the largest neuroimaging data of high-functioning ASD adult males in the ABIDE consortium. This rationale was stated in the first paragraph of the Method section (line xx- xx in p.x):

This recording site was chosen because their data had the largest number of high-functioning adult males with ASD.

However, we agree with the reviewer’s suggestion concerning reproducibility using independent datasets (see also our editorial responses and those to reviewer 1). Therefore, using two new datasets collected in the Indiana University and ETH Zürich, we confirmed that the original observations were qualitatively reproduced. These two datasets were chosen because apart from the original data, they had the largest or second largest sample size of high-functioning ASD adult males.

To clarify this, we added the following new paragraphs to the Results and Methods sections with four new supplementary figures (Supplementary Figs. 6-9) and two tables (Supplementary Figs. 2 and 3).

Results section (lines 9-21 in p. 11)

Reproducibility tests

We confirmed the reproducibility of these observations using two independent datasets collected in Indiana University (Supplementary Table 2) and ETH Zürich (Supplementary Table 3).

Analysis of both datasets yielded qualitatively the same hierarchal structures of the energy landscapes, consisting of the same six brain states (Supplementary Figs. 6b and 8b). In addition, significant differences in the major/intermediate state frequency between TD and ASD groups were also reproduced (Supplementary Figs. 6c and 8c). Moreover, we could identify the atypically lower indirect transition frequency and aberrantly longer duration of the major states in the ASD groups (Supplementary Figs. 6d-g, and 8d-g). Finally, we confirmed that the correlations between brain dynamics and behavioural indices were reproduced (Supplementary Figs. 6h-j and 8h-j), and the associations between these brain dynamics and across-network functional coordination were also replicated (Supplementary Figs. 7 and 9).

Methods section (lines 12-24 in p. 23)

Reproducibility test

We tested the reproducibility of the current results using two other datasets in ABIDE²⁹: data recorded in Indiana University (9 ASD and 12 TD individuals; TR 0.813s, TE 28ms,

Flip angle 60° for fMRI data; Supplementary Table 2) and those collected in ETH Zürich (10 ASD and 15 TD individuals; TR 2s, TE 25ms, Flip angle 90° for fMRI data; Supplementary Table 3). These two datasets were chosen because except the data of the University of Utah, they had the largest or second largest MRI images of ASD adult males in the ABIDE database.

We first selected high-functioning ASD adult males and age-/sex-/IQ-matched TD individuals based on the same criteria as those applied to the data of the University of Utah. We then applied the same analyses to these data, and examined the robustness of the current observations.

Concern 9

“8. (p. 15) Why 214 ROIs? The paper the authors reference includes 264.”

We are sorry for our insufficient descriptions concerning the ROI definition. Because the current study focused on dynamics of cortical brain activity, we excluded 50 ROIs whose network names were defined as ‘Uncertain’ or ‘Subcortical’ in the previous studies (Power et al., 2011; Cole et al., 2013).

To clarify this issue, we have now added the following statements to the Methods section (lines 18-20 in p. 17).

We then extracted a time series of fMRI signals from each of 214 regions of interest (ROIs) that were selected from the 264 ROIs listed in the previous studies^{30,31}. The other 50 ROIs were not adopted here because they were labelled ‘Uncertain’ or ‘Subcortical’ and did not constitute specific cortical networks.

References

30. Power, J. D. *et al.* Functional network organization of the human brain. *Neuron* **72**, 665–678 (2011).
31. Cole, M. W. *et al.* Multi-task connectivity reveals flexible hubs for adaptive task control. *Nature Neuroscience* **16**, 1348–1355 (2013).

Concern 10

“9. (p. 15) I would expect that cognitive control and attentional networks play distinct roles in social disorders like ASD. I recognize the convenience of combining them for computational reasons, but is this actually justified? It would be good if the authors could either demonstrate that these three systems independently follow similar time series (thus justifying the merger) or reproduce their analyses by adding either one (split ATN but retain merger of attention networks) or two (split ATN into three components) new nodes.”

We appreciate the reviewer raising this important issue. According to the reviewer’s suggestion, we examined the similarity between the binarised network activity in the nine cortical brain systems in which the DAN, VAN, and CON were not merged (Supplementary Figs. 11a and 11b). This analysis revealed that the similarity between the three networks was significantly higher than chance level (50%) for both the TD and ASD groups ($\geq 71\%$ in TD, $\geq 73\%$ in ASD, $P \leq 10^{-5}$ in one-sample *t*-tests

across participants). In contrast, the similarity between the other pairs of brain systems did not show such high similarity (<58%, $P > 0.05$). These results suggest that the activity of the three systems were significantly similar to each other even in ASD individuals.

In addition, according to the reviewer's suggestion, we examined the reproducibility of the results when the ATN was divided into the CON and the other two (i.e., DAN and VAN) (Supplementary Figs. 11c and 11d), and found qualitatively the same energy landscapes with the same six local minima. Notably, CON showed the same activity patterns as the DAN+VAN did.

These additional results provide empirical support for our merging of brain activity from the three systems into one time series. We therefore clarified this issue by now adding the following descriptions to the Methods section (lines 2-11 in p. 18) with a new supplementary figure (Supplementary Fig. 11).

The dorsal/ventral attention networks (DAN/VAN) and cingulo-opercular network (CON) were merged into the attention network (ATN), because (i) the current data size is not enough to perform energy-landscape analysis with nine factors and (ii) these three networks are considered to be responsible for the similar attention-related cognitive activity³⁰. In fact, in the binary form, brain activity patterns of these three networks were significantly similar to each other ($\geq 71\%$ in TD, $\geq 73\%$ in ASD, $P \leq 10^{-5}$ in one-sample t -tests across participants; Supplementary Figs. 11a and 11b), whereas those of the other networks did not show such high similarity (<58%). In addition, even after we divided the ATN into CON and the other two systems (DAN/VAN), we observed qualitatively the same energy landscapes with the same six local minima (Supplementary Figs. 11c and 11d). These results are considered to justify our merging the three networks into one system.

Supplementary Fig. 11

Concern 11

“10. (p. 15) The authors note that activity was binarized using the “average brain activity value”. They reference several papers but provide no further explanation as to what this entails. It would be good to be didactic in this explanation. I am also curious, then, about the robustness of their findings with respect to alternative thresholds (as the technique for selecting the threshold they reference is one of many and, in and of itself, is not privileged in any real way).”

According to the reviewer’s suggestion, we now added the following descriptions concerning the binarisation process and its purpose into the Methods section (lines 26- 31 in p. 17).

Technically, in each participant, we first calculated the average brain activity for each network, and binarised the original brain activity using the average activity as a threshold. ... This binarisation process enabled us to balance the number of active state and that of inactive state for each network, which could reduce the risk of overfitting in the following analysis⁶².

In addition, we confirmed the robustness of the current results against changing the binarisation thresholds (Supplementary Fig. 2): we could see qualitatively the same energy landscapes with the same six local minima when we set the threshold at -0.1 or 0.1 .

Such a high accuracy of model fitting and the hierarchal structures of energy landscapes were preserved when we changed the threshold for the binarisation of brain activity (Supplementary Fig. 2).

Supplementary Fig. 2

Concern 12

“11. (p. 19) Another possible concern is that the segregation of systems (which the authors define as the different of within - between module connectivity) could be influenced by global connectivity levels. It would be good to show that the

segregation measure is not significantly correlated across subjects with any of (A) sum of absolute value of all connections, (B) sum of all connections, (C), sum of only positive connections, or (D) sum only negative connections. If any of these correlations are strong, it suggests that the segregation measure may, in fact, be driven by network-wide connectivity differences rather than connectivity differences specific to the systems in question. □I hope that the authors find these comments useful.”

We acknowledge the reviewer’s concern about the effects of global connectivity levels. Therefore, according to the reviewer’s suggestion, we examined the correlations between functional segregation strengths and four types of global connectivity level, but found no significant correlations ($|r| \leq 0.16$) in both the TD and ASD groups.

We clarified this issue by adding the following description into the Methods section (lines 7-10 in p. 23) with a new supplementary table (Supplementary Table 4).

Note that in both TD and ASD groups, the functional segregation strength was not significantly correlated with global connectivity levels ($|r| \leq 0.16$, $P \geq 0.42$, Supplementary Table 4), which suggests that observations about functional segregation cannot be explained simply by global network-wide connectivity.

Supplementary Table 4

Correlation coefficients (r) between functional segregation strength and global FC level.

	Sum of FC	Sum of FCs	Sum of positive FCs	Sum of negative FCs
TD				
Functional segregation in Major states	0.082	-0.16	-0.021	-0.16
Functional segregation in Intermediate state	-0.075	0.040	-0.039	0.088
ASD				
Functional segregation in Major states	0.13	-0.0066	0.15	-0.11
Functional segregation in Intermediate state	0.073	-0.040	0.06	-0.072

Sum of |FC|: summation of the absolute values of FCs.

Sum of FCs: summation of FCs.

Sum of positive FCs: summation of only positive FCs.

Sum of negative FCs: summation of only negative FCs.

All of these summations were performed using Fisher-transformed FCs (i.e., Z value).

We thank the reviewer for their helpful comments which we believe have substantially improved our paper.

Responses to the comments of the editor and reviewers for “Brain network dynamics in high-functioning individuals with autism” by Watanabe and Rees (NCOMMS-16-26411).

Responses to reviewer #3’s comments

We appreciate the reviewer’s encouraging and thoughtful comments. We replied to all the reviewer’s comments in the same order as used by the reviewer. The page numbers refer to those in the revised manuscript unless otherwise mentioned. The modified or added text is in **blue** in the revised manuscript.

Concern 1

“This is a minor point, but I believe it’s more standard to refer to people who have a diagnosis of autism as individuals with autism or ASD, and their brains as the brains of individuals with autism/ASD (or phrases similar to that), as opposed to “autistic” individuals or brains.”

We appreciate the reviewer’s suggestion. We have now modified all the relevant phraseology.

Concern 2

“A relevant article that you did not cite is: Uddin, Supekar, Lynch, Cheng, Odriozola, Barth, Phillips, Feinstein, Abrams & Menon (2015). Brain state differentiation and behavioral inflexibility in autism. *Cerebral Cortex*, 25, 4740-7. It is one of the few articles to date that has measured temporal changes in functional connectivity in autism, and supports your findings that brain states are too stable in autism.”

Thank you - we have now added the following text citing these important findings.

Introduction section (lines 14- 15 in p. 3)

Although a recent study has reported atypical temporal interactions between different brain networks in individuals with autism and associated them with their aberrant behavioural inflexibility²⁵, how whole-brain neural activity patterns change over time in individuals with ASD is still poorly understood and thus, relationships between such brain dynamics and ASD symptoms are little identified.

Discussion section (lines 28-30 in p. 12)

Exceptionally, a recent fMRI study using Granger causality analysis has investigated patterns of temporal interactions between different brain regions and reported atypically stable temporal changes in functional connectivity²⁵.

Concern 3

“Did you look at data for any of the other ABIDE sites? Given that you chose to only look at a single site because of potential across-site differences and confounds, it would be useful to be able to demonstrate that these results are not a quirk of the single site you chose. Relating Utah results to results from one of the other sites

would be informative.”

We agree that it is important to reproduce our findings with independent datasets. Therefore, according to the reviewer’s suggestion (see also reviewer 1 and editorial suggestions), we have repeated all our analyses using two new datasets collected in Indiana University (Supplementary Table 2) and ETH Zürich (Supplementary Table 3), and confirmed that the original observations were qualitatively reproduced (Supplementary Figs. 6-9).

To report these new analyses, we have now added the following descriptions into the Results and Methods sections with the four new supplementary figures (Supplementary Figs. 6-9) and two tables (Supplementary Tables 2 and 3).

Results section (lines 9-21 in p. 11)

Reproducibility tests

We confirmed the reproducibility of these observations using two independent datasets collected in Indiana University (Supplementary Table 2) and ETH Zürich (Supplementary Table 3).

Analysis of both datasets yielded qualitatively the same hierarchal structures of the energy landscapes, consisting of the same six brain states (Supplementary Figs. 6b and 8b). In addition, significant differences in the major/intermediate state frequency between TD and ASD groups were also reproduced (Supplementary Figs. 6c and 8c). Moreover, we could identify the atypically lower indirect transition frequency and aberrantly longer duration of the major states in the ASD groups (Supplementary Figs. 6d-g, and 8d-g). Finally, we confirmed that the correlations between brain dynamics and behavioural indices were reproduced (Supplementary Figs. 6h-j and 8h-j), and the associations between these brain dynamics and across-network functional coordination were also replicated (Supplementary Figs. 7 and 9).

Methods section (lines 12-24 in p. 23)

Reproducibility test

We tested the reproducibility of the current results using two other datasets in ABIDE²⁹: data recorded in Indiana University (9 ASD and 12 TD individuals; TR 0.813s, TE 28ms, Flip angle 60° for fMRI data; Supplementary Table 2) and those collected in ETH Zürich (10 ASD and 15 TD individuals; TR 2s, TE 25ms, Flip angle 90° for fMRI data; Supplementary Table 3). These two datasets were chosen because except the data of the University of Utah, they had the largest or second largest MRI images of ASD adult males in the ABIDE database.

We first selected high-functioning ASD adult males and age-/sex-/IQ-matched TD individuals based on the same criteria as those applied to the data of the University of Utah. We then applied the same analyses to these data, and examined the robustness of the current observations.

Concern 4

“In Table 1, why not include p-values comparing ADOS scores across TD and ASD groups? It would be a nice way to emphasize the expected difference in scores.”

According to the reviewer’s suggestion, we have added *P* values for the ADOS scores into the revised Table 1. Note that the statistical values were computed based on 24 ASD and 16 TD individuals for whom ADOS scores were provided in the dataset.

Concern 5

“On page 4, line 113, you write “Coincidentally, we identified the same six locally stable brain activity patterns in the TD and ASD individuals...” Why do you think this is a coincidence? You write that in the Figure 2 caption as well. This finding is consistent with the work from Vince Calhoun’s lab demonstrating that different populations (i.e., schizophrenia versus healthy control subjects) have the same brain states, but different dwell time and transition patterns across brain states. Most likely, it is meaningful that individuals with autism have the same brain activity patterns as healthy individual, but different transition patterns and dynamics. This should be written about in the discussion as well.”

We appreciate the reviewer’s important point. We have now added the following paragraph concerning this issue into the Discussion section, citing relevant studies from Calhoun’s lab (lines 4-10 in p. 13).

Such a critical link between symptoms and brain dynamics is not limited to autism, but has been reported in recent human fMRI studies on schizophrenia^{47,48}. For example, one of these studies found that patients with schizophrenia and healthy controls showed similar static brain states, but exhibited significant differences in the dwell time in specific brain states and transition frequencies between such brain states⁴⁷. Given such prior observations, the current study can be seen as additional empirical support that highlights the importance of investigating brain dynamics in biological understanding of various developmental and psychiatric disorders^{11,49}.

References

11. Uhlhaas, P. J. & Singer, W. Neuronal Dynamics and Neuropsychiatric Disorders: Toward a Translational Paradigm for Dysfunctional Large-Scale Networks. *Neuron* **75**, 963–980 (2012).
47. Du, Y. *et al.* Interaction among subsystems within default mode network diminished in schizophrenia patients: A dynamic connectivity approach. *Schizophr. Res.* **170**, 55–65 (2016).
48. Ma, S., Calhoun, V. D., Phlypo, R. & Adalı, T. Dynamic changes of spatial functional network connectivity in healthy individuals and schizophrenia patients using independent vector analysis. *NeuroImage* **90**, 196–206 (2014).
49. Calhoun, V. D., Miller, R., Pearlson, G. & Adalı, T. The chronnectome: time-varying connectivity networks as the next frontier in fMRI data discovery. *Neuron* **84**, 262–274 (2014).

Concern 6

“Overall, your figures are very busy and therefore hard to follow. Perhaps separating them into more figures, spreading the different plots out more, shortening the axis titles, and adding legends so you do not have to have words written over the plots, would help.”

We are sorry for that the figures in our original manuscript were hard to follow. According to the reviewer's suggestion, we have now modified all the figures. In particular, we have simplified the original Figure 3. Regarding the original Figure 4, we have reorganised it and divided it into a new Figure 4 and Figure 5.

Concern 7

“Page 7, line 199: “...whereas such difference was seen in duration of Major states ($P > 0.56$).” I do not understand what this phrase means, please elaborate in the text. In that same paragraph, what is the ADOS score you are correlating? Overall score? One of the subscales? Please specify.”

Regarding the expression, “...whereas such difference was seen in duration of Major states ($P > 0.56$)”, we are sorry for the unclear description in the original manuscript. Now we have modified the expression as follows (lines 23-25 in p. 7):

..., whilst the duration of the major states was not significantly different between the TD individuals with higher and lower ADOS scores ($P > 0.56$).

Regarding the ADOS scores, the scores referred to the ADOS overall score. We have now clarified this issue by modifying the expression as follows (lines 18-23 in p. 7):

In the ASD group, the indirect transition frequency was negatively correlated with ADOS total scores ($r = -0.47$, $P_{\text{uncorrected}} = 0.01$, $P_{\text{Bonferroni}} < 0.05$; Fig. 4a), whereas the duration of the major states did not show a significant correlation ($r = -0.09$). Even in the TD group, the indirect transition frequency was significantly smaller in the neurotypical individuals with higher ADOS scores (ADOS total = 2–4) than in those with lower ADOS scores (ADOS total = 0–1) ($t_{16} = 2.6$, $P = 0.019$ in a two-sample t -test; Supplementary Fig. 3),

Concern 8

“The last sentence of that section: “These observations indicate that atypically unstable Intermediate state in autistic brains reduces their Indirect transition, which increases the severity of autistic symptoms (Figs. 4c).” is a bit strong, given that you do not know if the decreased indirect transition is what causes increased severity of ASD symptomatology. From this analysis you know that decreased indirect transitions are related to increased ASD symptomatology. Whether it is a direct cause or a hierarchical relationship caused by an unmeasured variable you do not know from this analysis. The same goes for the other results of similar analyses that you describe throughout the text.”

We agree. We have therefore toned down all of the relevant discussion throughout the entire manuscript. For example, the sentence the reviewer mentioned was re-written as follows (line 31 in p. 7 – line 2 in p. 8):

These observations indicate that the atypically unstable intermediate state in the brains of individuals with ASD is related to the reduction in the indirect transitions, and such

aberrant decreases in brain dynamics flexibility are associated with the severity of ASD symptoms.

Concern 9

“In Figure 4a, if you remove and/or deweight the outlier with the low ADOS score and the highest indirect transition frequency, does the reported relationship still hold?”

According to the reviewer’s suggestion, we re-analysed the correlation between ADOS and Indirect transition frequency after excluding the two outliers, and found that the correlation was still significant ($r = -0.46$, $P = 0.02$).

We have now added the following statement into the legend of Figure 4 (lines 5-6 in p. 31).

This correlation was preserved even after two outliers (squares circled by dashed lines) were removed ($r = -0.46$).

Concern 10

“Last line of page 8 “...and a module with the other four networks...” it would be helpful for the reader to list those four networks out. Same goes for the 5 networks mentioned on lines 279 and 296 on page 10.”

According to the reviewer’s suggestion, we have now modified the relevant expressions in the entire manuscript. For example, the expression raised by the reviewer was re-written as follows (line 8 in p. 9):

i.e., DMN/SMN/Auditory module and FPN/SAN/ATN/Visual module; Fig. 6a

Concern 11

“In Figures 6 and 7 (and the corresponding results section), did you also relate major state functional segregation to IQ in TD, and intermediate state functional segregation to ASD? It would be interesting to report those results.”

Thank you for these interesting suggestions. We have now calculated the correlation between functional segregation strength during the major states and FIQ in the TD individuals. We found, differently to the case of ASD individuals, that functional segregation strength showed a significantly negative correlation with FIQ in the TD data ($r = -0.44$, $P = 0.023$). This result adds further support for the notion that individuals with ASD have unique cognitive styles.

Second, we also investigated the association between functional segregation strength during the intermediate state and ADOS total scores in the TD data. We found that TD individuals with more ‘autistic’ ADOS scores (ADOS total = 2-4) showed less functional segregation during the Intermediate state than those with lower ADOS scores (ADOS total = 0-1). This observation is consistent with observations in the ASD data (Fig. 6).

We added the following descriptions about these results into the Results section with a new supplementary figure (Supplementary Fig. 5).

For the first point (lines 26-29 in p. 10):

In contrast, the correlation between FIQ and the functional segregation during the major states was not positive but significantly negative in the TD data ($r = -0.44$, $P = 0.023$; Supplementary Fig. 5b), which is consistent with previous reports suggesting that high-functioning individuals with ASD have different cognitive styles compared to TD individuals³⁸⁻⁴⁰.

For the second point (line 31 in p. 9 – line 2 in p. 10):

Such a significant association between ADOS scores and the functional segregation strength was observed even in the TD data ($t_{16} = 2.5$, $P = 0.025$ in a two-sample t -test, $P = 0.0001$ in a post-hoc permutation test, Cohen's $d = 1.4$; Supplementary Fig. 5a).

Supplementary Figure 5

Concern 12

“The discussion should be fleshed out a bit:

We appreciate the reviewer’s suggestions of important issues to discuss. For each particular issue, we have now added the following text to the Discussion:

1) What do you make of the different relationship between FIQ and brain states in TD and ASD? This is an interesting result and should be written about in the discussion.

Lines 8-17 in p. 14

In contrast, the general cognitive skill of the individuals with ASD was associated with the stability of their brain dynamics, not with its flexibility (Figs 4c and 5b), which could fit the unique cognitive style that high-functioning ASD individuals are supposed to have³⁸⁻⁴⁰. Behaviourally, high-functioning individuals with ASD are likely to show above-average performance when tasks they are engaged in require detail-focused information processing¹² not global one⁵⁵. This behavioural tendency well matches the overly stable brain dynamics observed in this study, if as suggested in a previous study⁸,

the stability of brain dynamics could be related to one's capability of repeating the same cognitive process. Although future studies need to examine the relationships between the stable brain dynamics and efficiency of information processing, the current findings may become a new foundation for biological understanding autism-specific cognitive styles.

“2) If you were to look at connectivity patterns across the 7 systems that describe each of these states, how do these patterns relate to what we know about network organization in ASD based on previous literature?”

Lines 12-23 in p. 13

The atypical across-network functional coordination that we observed in the ASD group is consistent with previous observations of atypical across-ROI FCs in the brains of individuals with autism⁵⁰⁻⁵². For example, a previous resting-state fMRI study reported atypical reduction in FC between the amygdala, which is often included in SAN, and secondary visual area⁵⁰. If this observation indicates a weak FC between SAN and the visual network, it matches the current findings about weak segregation during the intermediate state (Fig. 6) and strong segregation during the major states in the ASD data (Fig. 7). In the same logic, the current results are consistent with another resting-state fMRI study⁵¹ reporting an atypical decrease in the FC between a temporal region (Auditory network) and a medial prefrontal area (DMN) in autism. We can also see consistency in a task-based fMRI study that found weak functional coupling between a visual area and a region in FPN in high-functioning adults with ASD⁵². Although more research is needed, such consistencies between the present results and previous findings provide some reassurance concerning the current findings about across-network functional coordination in the brains of autism.

“3) Why is it that increased segregation during intermediate states and decreased segregation during major states is more typical/relevant to healthy cognition and brain dynamics?”

Line 25 in p. 13 – line 6 in p. 14

The current study has also identified brain dynamics associated with the general cognitive ability in neurotypical adults (Fig. 5a). Cognitive skills in the TD participants were positively associated with the flexibility of brain dynamics, and such flexible brain dynamics were underpinned by the increased functional segregation during the intermediate state (Fig. 8) and the decreased functional segregation during the major states (Supplementary Fig. 5b). This functional coordination during the major and intermediate states may enable the control of diverse cognitive functions in an integrative manner. Theoretically, smooth integration of functionally different brain systems is considered to be vital for binding diverse perceptual information and achieving better cognitive performance in a changing environment^{4,6,37,53}. Empirically, several neuroimaging studies have suggested that such an integration process is achieved by frequent transitions between different brain states^{8,54}. Considering the current results with these theoretical and empirical observations, we can speculate that the functional coordination seen in the neurotypical major and intermediate states may contribute to integrative information processing by facilitating transitions between different brain activity states.

References

4. Deco, G., Tononi, G., Boly, M. & Kringelbach, M. L. Rethinking segregation and integration: contributions of whole-brain modelling. *Nat. Rev. Neurosci.* **16**, 430–439 (2015).
6. Breakspear, M. Dynamic models of large-scale brain activity. *Nature Neuroscience* **20**, 340–352 (2017).
8. Shine, J. M. *et al.* The Dynamics of Functional Brain Networks: Integrated Network States during Cognitive Task Performance. *Neuron* **92**, 544–554 (2016).
12. Happé, F. & Frith, U. The weak coherence account: detail-focused cognitive style in autism spectrum disorders. *J Autism Dev Disord* **36**, 5–25 (2006).
37. Schultz, D. H. & Cole, M. W. Integrated Brain Network Architecture Supports Cognitive Task Performance. *Neuron* **92**, 278–279 (2016).
38. Happe, F. Autism: cognitive deficit or cognitive style? *Trends in Cognitive Sciences* **3**, 216–222 (1999).
39. Shah, A. & Frith, U. Why do autistic individuals show superior performance on the block design task? *J Child Psychol & Psychiat* **34**, 1351–1364 (1993).
40. Dawson, M., Soulières, I., Gernsbacher, M. A. & Mottron, L. The level and nature of autistic intelligence. *Psychol Sci* **18**, 657–662 (2007).
50. Rudie, J. D. *et al.* Reduced functional integration and segregation of distributed neural systems underlying social and emotional information processing in autism spectrum disorders. *Cereb. Cortex* **22**, 1025–1037 (2012).
51. Abrams, D. A. *et al.* Underconnectivity between voice-selective cortex and reward circuitry in children with autism. *Proc. Natl. Acad. Sci. U.S.A.* **110**, 12060–12065 (2013).
52. Koshino, H. *et al.* fMRI investigation of working memory for faces in autism: visual coding and underconnectivity with frontal areas. *Cereb. Cortex* **18**, 289–300 (2008).
53. Rabinovich, M. I., Leekam, S. R., Prior, M. R., Varona, P. & Uljarevic, M. Robust transient dynamics and brain functions. *Front Comput Neurosci* **5**, 24 (2011).
54. Hellyer, P. J., Scott, G., Shanahan, M., Sharp, D. J. & Leech, R. Cognitive Flexibility through Metastable Neural Dynamics Is Disrupted by Damage to the Structural Connectome. *J. Neurosci.* **35**, 9050–9063 (2015).
55. Booth, R. D. L. & Happé, F. G. E. Evidence of Reduced Global Processing in Autism Spectrum Disorder. *J Autism Dev Disord* (2016).

Concern 13

“In the Data Preprocessing section of the methods (page 15), you can quantify if the timeseries of the DAN, VAN, and CO networks are similar enough to justify averaging together. Please report that.”

According to the reviewer’s suggestion, we examined the similarity in the binarised brain activity between the three networks. As a control, we also calculated the similarity between the other pairs of the nine cortical brain networks listed in the previous literature (Power *et al.*, 2011; Cole *et al.*, 2013). We found that the similarity between the three networks was significantly higher than chance level (50%) in both the TD and ASD groups ($\geq 71\%$ in TD, $\geq 73\%$ in ASD, $P \leq 10^{-5}$ in one-sample *t*-tests across participants). In contrast, the similarity between the other pairs of brain systems did not show such high similarity ($< 58\%$, $P > 0.05$). These results suggest that the activity of the three systems were significantly similar to each other even in ASD individuals.

These additional results provide empirical support for our merging of brain activity from the three systems into one time series. We have reported these findings by adding the following descriptions

to the Methods section (lines 2-11 in p. 18) with a new supplementary figure (Supplementary Fig. 11).

The dorsal/ventral attention networks (DAN/VAN) and cingulo-opercular network (CON) were merged into the attention network (ATN), because (i) the current data size is not enough to perform energy-landscape analysis with nine factors and (ii) these three networks are considered to be responsible for the similar attention-related cognitive activity³⁰. In fact, in the binary form, brain activity patterns of these three networks were significantly similar to each other ($\geq 71\%$ in TD, $\geq 73\%$ in ASD, $P \leq 10^{-5}$ in one-sample t -tests across participants; Supplementary Figs. 11a and 11b), whereas those of the other networks did not show such high similarity ($< 58\%$). In addition, even after we divided the ATN into CON and the other two systems (DAN/VAN), we observed qualitatively the same energy landscapes with the same six local minima (Supplementary Figs. 11c and 11d). These results are considered to justify our merging the three networks into one system.

References

30. Power, J. D. *et al.* Functional network organization of the human brain. *Neuron* **72**, 665–678 (2011).
31. Cole, M. W. *et al.* Multi-task connectivity reveals flexible hubs for adaptive task control. *Nature Neuroscience* **16**, 1348–1355 (2013).

We thank the reviewer for their helpful comments which we believe have substantially improved our paper.

Reviewers' comments:

Reviewer #1 (Remarks to the Author):

The authors have done a nice job of replicating original findings and responding to all previous comments. This manuscript will make a significant and novel contribution to the autism literature.

Reviewer #2 (Remarks to the Author):

This review is of a revised manuscript, the previous version of which I had also reviewed. In their revision, the authors perform additional analyses, including reproducing their results (at least qualitatively) using additional datasets. They also go on to perform some, but not all, of the tests that I suggested. Overall, I only have one real concern and, provided that the authors can address it, I would be happy to recommend the paper for publication. Otherwise they addressed all of my other concerns.

Major Concern:

In my original review I noted, and the authors seem to acknowledge, that the MEM approach was limited by the number of regions between which connection weights can be estimated. They go on to note that other studies have employed similar system-level analysis of the BOLD signal, which I cannot dispute. With that said, it is worth acknowledging that there are inconsistencies across studies as to how the systems, themselves, should be defined. For example, the study by Thomas Yeo identifies only one control network (Yeo et al, 2011) while Jonathan Power's study (approximately) sub-divides the control network into cingulo-opercular and fronto-parietal systems (Power et al, 2011). Other studies have even argued that the default mode network ought to be sub-divided into distinct sub-systems (Uddin et al 2009).

All of this to say that (1) the small number of nodes is a limitation and (2) the definition of nodes based on systems reported in the Power paper is not particularly privileged. I think that it would still be advisable to either repeat the analysis using a different set of sub-systems or with a finer-grained division of the current set of systems (to the extent that it is computationally tractable). The decision of how to define nodes has implications on the organization of connections among those nodes (Zalesky et al 2010).

Minor concern:

In measuring the similarity of system and regional/voxel time series, what measure of similarity was used? The authors should report this. And are these similarity values calculated based on the binarized time series of the BOLD signal?

References:

[1] Jonathan D Power, Alexander L Cohen, Steven M Nelson, Gagan S Wig, Kelly Anne Barnes, Jessica A Church, Alecia C Vogel, Timothy O Laumann, Fran M Miezin, Bradley L Schlaggar, et al. Functional network organization of the human brain. *Neuron*,

72(4):665{678, 2011.

[2] Lucina Q Uddin, AM Clare Kelly, Bharat B Biswal, F Xavier Castellanos, and Michael P Milham. Functional connectivity of default mode network components: correlation, anticorrelation, and causality. *Human brain mapping*, 30(2):625{637, 2009.

[3] BT Thomas Yeo, Fenna M Krienen, Jorge Sepulcre, Mert R Sabuncu, Danial Lashkari, Marisa Hollinshead, Joshua L Roman, Jordan W Smoller, Lilla Zöllei, Jonathan R Polimeni, et al. The organization of the human cerebral cortex estimated by intrinsic functional connectivity. *Journal of neurophysiology*, 106(3):1125{1165, 2011.

[4] Andrew Zalesky, Alex Fornito, Ian H Harding, Luca Cocchi, Murat Yuucel, Christos Pantelis, and Edward T Bullmore. Whole-brain anatomical networks: does the choice of nodes matter? *Neuroimage*, 50(3):970{983, 2010.

I hope that the authors find these comments useful.
Richard Betzel

Reviewer #3 (Remarks to the Author):

The authors responded thoroughly to the reviewers' comments and have made this manuscript a much stronger contribution to the literature. I have no other comments of substance.

One small comment is about Figure 8b. In the text you write: "In contrast, the correlation between FIQ and the functional segregation during the major states was not positive but significantly negative in the TD data ($r = -0.44$, $P = 0.023$; Supplementary Fig. 5b), which is consistent with previous reports suggesting that high-functioning individuals with ASD have different cognitive styles compared to TD individuals." (last sentence, page 9). But Figure 8b, third plot seems to report that functional segregation is positively correlated with FIQ ($r = 0.46$, $p = .018$). Is the difference that it is functional segregation during the immediate state in that plot? If so, it would be more clear if the x-axis of Figures 7 and 8 specify of what state the Functional segregation is measured.

I also noticed a few typos and missed words in the added text.

Responses to the comments of the editor and reviewers for “Brain network dynamics in high-functioning individuals with autism” by Watanabe and Rees (NCOMMS-16-26411A).

Responses to the reviewer #2’s comments

We appreciate the reviewer’s useful suggestions. We have addressed all the comments of the reviewer. Page numbers in our responses refer to those in the revised manuscript unless otherwise mentioned. The modified or added text is in **blue** in the revised manuscript.

Major concern

“In my original review I noted, and the authors seem to acknowledge, that the MEM approach was limited by the number of regions between which connection weights can be estimated. They go on to note that other studies have employed similar system-level analysis of the BOLD signal, which I cannot dispute. With that said, it is worth acknowledging that there are inconsistencies across studies as to how the systems, themselves, should be defined. For example, the study by Thomas Yeo identifies only one control network (Yeo et al, 2011) while Jonathan Power's study (approximately) sub-divides the control network into cingulo-opercular and fronto-parietal systems (Power et al, 2011). Other studies have even argued that the default mode network ought to be sub-divided into distinct sub-systems (Uddin et al 2009).

All of this to say that (1) the small number of nodes is a limitation and (2) the definition of nodes based on systems reported in the Power paper is not particularly privileged. I think that it would still be advisable to either repeat the analysis using a different set of sub-systems or with a finer-grained division of the current set of systems (to the extent that it is computational tractable). The decision of how to define nodes has implications on the organization of connections among those nodes (Zalesky et al 2010).”

We agree with the importance of examining whether the current observations are reproduced using different brain parcellation methods. Therefore, according to the reviewer’s suggestion, we have now performed two additional analyses using two different brain parcellation methods, and confirmed the qualitative reproducibility of all the current findings in both cases (Supplementary Figs. 10 and 11).

One of the additional parcellation methods was based on the previous resting-state fMRI study (Uddin et al., 2009). According to this literature, we divided the DMN into two subnetworks by classifying the DMN ROIs into two groups: ROIs whose activities were mainly correlated with that of ventromedial prefrontal cortex (vmPFC) and ROIs whose activities were mainly correlated with that of the posterior cingulate cortex (PCC).

Technically, for each ROI of the DMN, we calculated Pearson correlation coefficients between the brain activity of the DMN ROI and those of vmPFC and PCC. After averaging them across participants, we compared the correlation with vmPFC to that with PCC, and categorised the DMN ROI: if the correlation with vmPFC was larger than that with PCC, the ROI was classified into vmPFC-DMN. Otherwise, the ROI was labelled as a region of PCC-DMN. The other six networks were the same as those in the original analysis, and eight brain networks were determined in total. Based on this network definition, we have repeated the same energy-landscape analyses, and confirmed that all the current observations were qualitatively reproduced (Supplementary Fig. 10). Note that we could not see differences in activity patterns between the two DMN subnetworks.

The other brain parcellation method was based on another previous study the reviewer kindly raised (Yeo et al., 2011). Using a “7 network tight mask” proposed in the study, we parcellated the cortical area into seven brain systems that were different from those adopted in the original analyses. We then extracted average neural activity for each brain system, and repeated the entire analyses. Consequently, we have observed that brain dynamics of individuals with ASD were more stable than those of the controls and such neural stability was correlated with the ASD severity and the cognitive skills of the individuals with ASD (Supplementary Fig. 11).

In addition, given that the Limbic system in this new parcellation method overlaps with the SAN in the original analyses, and the SM system almost includes the original Auditory network, the newly-obtained energy landscapes and their local minima appear qualitatively the same as those in the original analyses (see legends of Supplementary Fig. 11 for details). In this sense, the results based on this additional parcellation method could be seen as a qualitative replication of the original findings.

We have reported these results by adding the following descriptions into the Results and Methods section with two new supplementary figures (Supplementary Figs. 10 and 11).

Results section (line 21 in p.11 – line 3 in p.12):

We also tested whether the current observations were robust against differences in the definitions of brain networks. To this end, we repeated the energy-landscape analyses after the brain was parcellated in two different manners^{40,41}.

In one of the brain parcellation methods, the DMN was divided into two sub-networks according to a previous study⁴⁰ (see Methods for details). Although the accuracy of the model fitting was slightly reduced (82.1% for TD and 80.7% for ASD), we observed qualitatively the same energy landscapes, brain dynamics, and brain-behaviour associations as seen in the original analyses (Supplementary Fig. 10).

In the other brain network definition, the cortical area was parcellated into a different set of seven brain systems based on another previous study⁴¹ (Supplementary Fig. 11a). Even in this brain division, we still found that the neural dynamics of individuals with ASD were more stable than those of the control (Supplementary Fig. 11c-g), and such neural stability showed positive correlations with the severity of their symptoms and their cognitive skills (Supplementary Fig. 11h-j).

Methods section (line 30 in p.24 – line 5 in p.26):

Reproducibility tests: different brain parcellation methods

We also tested the robustness of the current observations using two different definitions of brain networks, because choices of brain parcellation methods could affect results of some calculations about large-scale brain architecture⁷⁰.

In one of the new brain parcellation methods, which was a finer version of the original brain network definition, we divided the DMN into two subnetworks according to a previous study⁴⁰. Technically, we classified the DMN ROIs into two groups: ROIs whose activities were mainly correlated with that of ventromedial prefrontal cortex (vmPFC, [2,

54, -3] in Talairach coordinates) and ROIs whose activities were mainly correlated with that of posterior cingulate cortex (PCC, [-2, -51, 27] in Talairach coordinates). The vmPFC and PCC were chosen because the previous study showed their distinct roles in whole-brain network coordination⁴⁰. The anatomical coordinates of the regions were determined based on the study.

For example, we categorised an ROI_{*i*} in DMN as follows. (i) For each participant, we calculated two Pearson correlation coefficients between the brain activities of the ROI_{*i*}, vmPFC, and PCC (i.e., an ROI_{*i*}-vmPFC correlation and an ROI_{*i*}-PCC correlation). (ii) After applying Fisher's Z transformation to the correlation coefficients and averaging the Z-scores across all the participants, we compared the average ROI_{*i*}-vmPFC correlation with the average ROI_{*i*}-PCC correlation. (iii) If the ROI_{*i*}-vmPFC correlation was larger than the ROI_{*i*}-PCC correlation, the ROI_{*i*} was labelled as a region of vmPFC-DMN. Otherwise, the ROI_{*i*} was categorised as a region of PCC-DMN. (iv) We repeated this calculation for all the 59 ROIs of the DMN.

As a result of this procedure, we obtained a vmPFC-DMN with 27 ROIs and a PCC-DMN with 32 ROIs. The other six networks were the same as those in the original analysis. In total, we determined eight brain networks (vmPFC-DMN, PCC-DMN, FPN, SAN, ATN, SMN, Auditory, and Visual), and repeated the same energy-landscape analysis for the eight brain systems.

The other brain parcellation was adopted from a different previous study⁴¹. In the study, the cortical area was divided into seven brain systems that were not exactly the same as those used in the original analyses (Supplementary Fig. 11a). Technically, we estimated the average neural activity for each brain system using a "7 network tight mask" (surfer.nmr.mgh.harvard.edu/fswiki/CorticalParcellation_Yeo2011), and repeated the entire analysis. This previous study also proposed a 17-network brain parcellation method, but we did not choose it because the size of the current dataset was too small to accurately perform energy-landscape analysis for such 17 networks.

These robustness tests against differences in brain parcellation methods were conducted using the dataset collected at the University of Utah.

Minor concern

"In measuring the similarity of system and regional/voxel time series, what measure of similarity was used? The authors should report this. And are these similarity values calculated based on the binarized time series of the BOLD signal?"

We calculated the similarity scores based on the binarised neural activity data using the following equation, $(N_T - \|\mathbf{x}_i - \mathbf{x}_j\|^2) / N_T$, where \mathbf{x}_i were a vector representing a time series of a binarised neural activity of brain network i (or ROI_{*i*}), and N_T denoted the length of \mathbf{x}_i .

We clarified this issue by adding the following description into the Methods section:

Methods section (lines 10-21 in p.19):

The similarity between brain activity \mathbf{x}_i and \mathbf{x}_j was calculated as $(N_T - \|\mathbf{x}_i - \mathbf{x}_j\|^2)/N_T$, where \mathbf{x}_i were a vector representing a time series of the binarised neural activity of brain network i (or ROI $_i$), and N_T denoted the length of \mathbf{x}_i .

We thank the reviewer for their helpful comments which we believe have substantially improved our paper.

Responses to the comments of the editor and reviewers for “Brain network dynamics in high-functioning individuals with autism” by Watanabe and Rees (NCOMMS-16-26411A).

Responses to reviewer #3’s comments

We appreciate the reviewer’s helpful comments. We have replied to all the reviewer’s comments in the same order as used by the reviewer. The page numbers refer to those in the revised manuscript unless otherwise mentioned. The modified or added text is in **blue** in the revised manuscript.

Comment 1

“One small comment is about Figure 8b. In the text you write: “In contrast, the correlation between FIQ and the functional segregation during the major states was not positive but significantly negative in the TD data ($r = -0.44$, $P = 0.023$; Supplementary Fig. 5b), which is consistent with previous reports suggesting that high-functioning individuals with ASD have different cognitive styles compared to TD individuals.” (last sentence, page 9). But Figure 8b, third plot seems to report that functional segregation is positively correlated with FIQ ($r = 0.46$, $p = .018$). Is the difference that it is functional segregation during the immediate state in that plot? If so, it would be more clear if the x-axis of Figures 7 and 8 specify of what state the Functional segregation is measured.”

We are sorry for any lack of clarity in our figures. As the reviewer stated, the x-axis in Figure 8b (and 6d-6f) indicates the strength of the functional segregation for the intermediate state, whereas that in Figure 7d-7f denotes that for the major state.

We have now clarified this issue by adding descriptions about the x-axis into the figures as follows:

Fig. 6

Fig. 7

Fig. 8

Comment 2

“I also noticed a few typos and missed words in the added text.”

Thank you. We have now checked the manuscript thoroughly and corrected some typos (e.g., “high-functioning” in line 2 in p. 14).

We thank the reviewer for their helpful comments which we believe have substantially improved our paper.

REVIEWERS' COMMENTS:

Reviewer #2 (Remarks to the Author):

The authors have addressed my concerns. I am happy to recommend this manuscript for publication.